# Thermodynamic Relationships for Perfectly Elastic Solids Undergoing Steady-State Heat Flow

**DOI:** 10.3390/ma15072638

**Published:** 2022-04-03

**Authors:** Anne M. Hofmeister, Everett M. Criss, Robert E. Criss

**Affiliations:** 1Department of Earth and Planetary Science, Washington University, St. Louis, MO 63130, USA; criss@wustl.edu; 2H10 Capital, 2401 4th Avenue, Suite 480, Seattle, WA 98121, USA; everett@h10capital.com

**Keywords:** steady state, heat, flux, perfectly frictionless elastic solids, Young’s modulus, energy reservoirs, interatomic forces, heat capacity, bulk modulus, thermal expansivity

## Abstract

Available data on insulating, semiconducting, and metallic solids verify our new model that incorporates steady-state heat flow into a macroscopic, thermodynamic description of solids, with agreement being best for isotropic examples. Our model is based on: (1) mass and energy conservation; (2) Fourier’s law; (3) Stefan–Boltzmann’s law; and (4) rigidity, which is a large, yet heretofore neglected, energy reservoir with no counterpart in gases. To account for rigidity while neglecting dissipation, we consider the ideal, limiting case of a perfectly frictionless elastic solid (PFES) which does not generate heat from stress. Its equation-of-state is independent of the energetics, as in the historic model. We show that pressure-volume work (*P**dV*) in a PFES arises from internal interatomic forces, which are linked to Young’s modulus (Ξ) and a constant (*n*) accounting for cation coordination. Steady-state conditions are adiabatic since heat content (*Q*) is constant. Because average temperature is also constant and the thermal gradient is fixed in space, conditions are simultaneously isothermal: Under these dual restrictions, thermal transport properties do not enter into our analysis. We find that adiabatic and isothermal bulk moduli (*B*) are equal. Moreover, *Q*/*V* depends on temperature only. Distinguishing deformation from volume changes elucidates how solids thermally expand. These findings lead to simple descriptions of the two specific heats in solids: ∂ln(*c_P_*)/∂*P* = −1/*B*; *c_P_* = *n*Ξ times thermal expansivity divided by density; *c_P_* = c_V_*n*Ξ/*B*. Implications of our validated formulae are briefly covered.

## 1. Introduction

Classical thermodynamics is an important tool in the physical sciences and engineering. Nevertheless, the equations and postulates developed in the 1800s should be called “thermostatics”, since time-dependent behavior is not part of this historic model [1]. Yet, dynamic, evolutionary behavior is ubiquitous. The flow of heat and its radiation from the system of interest are integral components of real processes. Idealizations needed to avoid addressing dynamic behavior in thermostatics are connected with restrictive approximations. A key example is the concept of reversibility, which is still currently debated [2]. Perceived reversibility rests on restoring changes in a system at the expense of altering the surroundings, which are neglected in such assessments, e.g., [3].

The macroscopic theory of “thermostatics” predates a rudimentary understanding of atomic structure and acceptance of light and heat as being the same phenomenon [4]. These omissions are understandable as they pertain mainly to microscopic behavior. Stefan’s observation of heat flux linking to temperature and Fourier’s theory of heat transfer, both from the 1800s, were not considered. The last omission was a significant error [5] because Fourier’s depiction of heat, a key entity in thermostatics, is likewise macroscopic. Key aspects of both laws and their relevance to thermostatics are as follows:

First, Fourier assumed that as heat flows through a sample, some heat is stored in mass elements along its path, while another fraction moves from element to element during flow. Regarding the latter fraction, Fourier defined the key, dynamic quantity of flux (ℑ, heat per area per time) and related it to the temperature (*T*) gradient:(1)$$ℑ=-\kappa \nabla T,\;\mathrm{or}\;\mathrm{in}\;\mathrm{one}\;\mathrm{dimension}:ℑ=-\kappa (T,P)\frac{\partial T}{\partial L}=-\hat{L}\frac{\partial T}{\partial L}\kappa (T,P)$$
where *κ* is thermal conductivity, *P* is pressure, and volume, *V*, goes as *L*^3^ in an isotropic medium. The unit vector denotes the specific direction of heat flow. Because any matrix representation can be diagonalized, the one-dimensional Cartesian form on the right-hand side (RHS) embodies the physics of heat transport.

Equation (1) states that net heat flows down the thermal gradient, which is equivalent to rudimentary articulations of the 2nd law (e.g., [6]). For gas, a thermal gradient stratifies density, generating unopposed buoyancy forces that cause convection. Yet, Equation (1) shows that under time-independent circumstances, the thermal gradient in solids is a vector quantity that is completely established by the transport property *κ* and experimental (boundary) conditions. The rigidity of solids permits heat to flow from the hot to the cold end without the net momentum transport that is inherent to gases.

Second, flux is universally tied to temperature via Stefan–Boltzmann’s law, thereby linking a dynamic entity to a key variable which is presumed to be static in the classical model. Stefan showed experimentally circa 1872 that radiated flux from all frequencies of light from a graphite-coated metal filament per area per time is:(2)$$ℑ=\sigma_{SB}T^{4},$$
where the Stefan–Boltzmann constant, *σ_SB_* = 5.670 × 10^−8^ Wm^−2^K^−4^ describes a blackbody (see Section 2.1). Temperature is thus defined by heat loss to the surroundings. In classical thermostatics, *T* is related to heat content *Q*, but not in a simple way, e.g., [6].

Third, time (*t*) is an explicit variable in Fourier’s second equation, which is obtained by taking a spatial derivative of Equation (1) and conserving energy. In 3-dimensions:(3)$$\rho c_{P}\frac{\partial T}{\partial t}=\nabla \cdot \left(\kappa \nabla T\right),$$
where *ρ* is density and *c_P_* is specific heat (on a per mass basis). Thermal conductivity governs the thermal evolution of a system, embodying how much heat is flowing and how fast. When changes in *T* are small, Equation (3) simplifies to:(4)$$\frac{\partial T}{\partial t}=D\nabla^{2}T,\;\mathrm{or}\;\mathrm{in}\;\mathrm{one}\text{-}\mathrm{dimension}:\;\frac{\partial T}{\partial t}=D\frac{\partial^{2}T}{\partial L^{2}}.$$

Thermal diffusivity (*D*) is also a dynamic property, describing the rate at which *T* evolves, independent of the amount of heat that is flowing. By definition:(5)$$\frac{\kappa }{D}=\rho c_{P}=C.$$

The static properties of the middle term can be individually measured. Their product *C*, called storativity, describes heat capacity on a per volume basis. Its importance in Equation (3) stems from diffusion depending on length-scale [3,7].

Last, heat transfer under pressure involves the *P* dependence of specific heat. The classical equation: (6)$$\frac{\partial c_{P}}{\partial P}=-TV\left(\alpha^{2}+\frac{\partial \alpha }{\partial T}\right),\;\mathrm{historical}$$
depends on thermal expansivity, α ≡ *V*^−1^∂*V*/∂*T*. This historical identity thus portrays the response of a static property to compression (*P* on the left-hand side, LHS) as arising from changes caused by heating (*T* and α on the RHS). Yet, diverse observations show that solids respond to heating and to compression in different ways, as embodied in the quasi-harmonic model of solids [8]. In particular, examining accurate experimental measurements of the *P* dependence of *κ* for 20 different solids that also had accurate material properties suggests that ∂(ln*c_P_*)/∂*P* depends simply on the inverse of the isothermal bulk modulus, *B_T_* = −*V*(∂*V*/∂*P*)^−1^ [7].

### 1.1. Different Behaviors of Solids and Gases May Affect Thermostatic Equations

Gas behavior is of long-standing importance to basic physics. Because solids behave much differently than gases (Figure 1), the same equations need not apply to these two distinct states of matter.

Constructs for heat storage in gas and solids must each account for differences in the types of energy stored, plus restrictions on converting energy between the different reservoirs. Crucially, for solids, heat transfer is independent of mass diffusion, as shown by Hofmeister and Criss [9]. Heat may be stored in the cyclical and microscopically localized vibrations of interatomic bonds in solids, but its transport across the solid does not involve net displacement of the atoms or deformation of their structural arrangement. Moreover, the vibrations cannot be the main energy reservoir of the solid because these constitute perturbations of the atoms from their static positions. Geometrical constraints limit average displacements to circa interatomic distances. These behaviors stem from solid matter’s strength and hallmark characteristic of rigidity (Figure 1). Solids deforming under shear stress greatly contrasts with behavior of gases, which flow under any stress and in which heat moves with the translations of its molecules. Thus, energy in a solid is essentially potential (stored) energy, whereas much of the energy in a gas is kinetic (translational) energy. For monatomic gas, all energy is translational.

### 1.2. Purpose and Limitations of the Paper

The present paper derives new relationships among thermostatic variables and properties for solids by considering steady-state heat transfer, which involves variations of *T* with position, but not with time. Isotropic solids are the focus for simplicity, availability of data, and because these embody the physical principles. Perfectly frictionless elastic solids (PFES) which do not generate heat as a function of time during changes are consistent with diverse equation-of-state (EOS) formulations. These formulations do not specify the energy difference between different states, so they effectively neglect how work and/or heat change *V*, *P*, and/or *T*. Mass, charge, and energy are conserved in our analysis.

Our model is macroscopic. Macroscopic approaches can provide a simple description of things that can be measured or sensed, and require no special assumptions concerning the nature of matter, yet yield straightforward, testable predictions that can disclose theoretical connections between measurable quantities [10]. Validation is a key component of any such endeavor. In this report, validation is mostly limited to isotropic solids for simplicity and to focus on physical principles.

Modeling transport properties, which describe time-dependent interactions and moreover depend on the length-scale [3,7], is beyond the scope of the present paper. Static physical properties (e.g., specific heat, storativity, thermal expansivity) are investigated here. Bulk moduli are part of classical theory, but shear moduli (*G*) are not. We focus on the heretofore neglected elastic moduli because these are essential to describe the forces inside a solid and therefore its energetics.

### 1.3. Organization of the Paper and Key Results

Section 2.1 discusses the crucial connection of broadband thermal emissions with temperature. Section 2.2 specifies why steady-state conduction in solids constrains both adiabatic and isothermal responses. Section 2.3 covers the equation-of-state for an isotropic PFES and explains why describing work requires an additional property, namely Young’s modulus. Section 2.4 uses elastic properties of isotropic solids to derive formulae for the *P* and *T* dependencies of heat capacity and heat storage. Section 3 evaluates our formulae and historic formulae against experimental results, focusing on ambient conditions due to accuracy and availability of data. For the reader’s convenience, Table 1 lists new, useful formulae for solids and the sections where these were derived and confirmed. Section 4 summarizes key findings and discusses implications of our results for basic and applied sciences. Section 5 concludes.

## 2. Theoretical Description of Solids Conducting Heat in Steady State

### 2.1. Link of Temperature to Heat Flux

Temperature is a macroscopic property arising from the thermal energy of an object, which differs from, but is related to, its heat content (*Q*). The direct link between *T* and heat flux (2), historically established for solids, pertains to this complicated relationship.

In detail, total flux includes all emitted light, and is obtained by integrating the intensity (*I*) over frequency (*ν*):(7)$$ℑ(T)=4\pi \int_{0}^{\infty }I(\nu ,T)d\nu ,$$
where ℑ is measured over a spherical surface enclosing the emitting object.

Difficulties in measuring absolute intensity are well-known (e.g., Figure 2). Hence, idealized behavior of a perfectly absorbing blackbody (BB) has been the theoretical focus. Planck’s function for this unachievable idealization (for *ν* in Hertz) is:(8)IBB(ν,T)=2hν3c21exp(hνkBT)−1,
where *h* = Planck’s constant, *c* = lightspeed, and *k_B_* is Boltzmann’s constant.

Because all hot matter emits thermal radiation, Equation (7) omits the subscript BB. The simplest scenario approximating reality is that of a greybody where *I* = ξ*I_BB_* and emissivity (ξ) is independent of both *ν* and *T*. Metals and graphite were used in classic experiments (Figure 2) because these strongly absorb and have optical functions that vary slowly with *ν* and *T*. Transparent material (e.g., silicate glasses) also have emissions, but these are related to *I_BB_* in a complicated manner that depends on the size of the object, absorption characteristics, surface reflections, and thermal gradients [11]. Gases are extremely transparent and were historically considered not to emit.

**Figure 2 materials-15-02638-f002:**
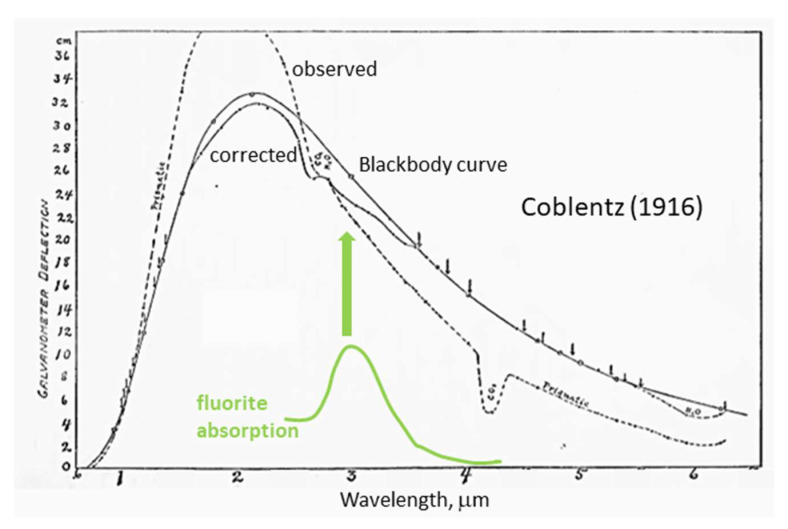
Emission curves of cavity radiation at 1370 K from Coblentz [12] compared to a near-IR absorption spectrum of natural fluorite (green curve, with an arbitrary y-scale). Dashed line = raw data, labeled “prismatic”. Solid curve with small dots = corrected data. Solid line with circles = the ideal Planck curve. Arrows indicate points Coblentz [12] used to fit the blackbody curve and determine the maximum. He omitted regions connected with atmospheric absorptions, in which features are partly due to use of natural fluorite as a prism, and in which material contains impurity bands.

#### 2.1.1. Wien’s Law

Wien’s historical experiments showed that the peak wavelength (*λ*) for a greybody is inversely proportional to *T*:(9)$$\lambda_{peak}=\frac{b}{T}\;\mathrm{or}\;\nu_{peak}=\frac{w_{3}k_{B}}{h}T\;\mathrm{where}\;\lambda_{peak}\neq \frac{c}{\nu_{peak}}$$
where *b* = 2897.8 μm K^−1^ was experimentally determined. The irrational number *w*_3_ (~2.821439) on the RHS was derived from *I_BB_* (Equation (8)) by numerically solving a transcendental equation [13,14]. Thus, ascertaining *T* from Equation (9) implicitly assumes a broad and skewed spectrum of a greybody (Figure 2), whereby *ν*_max_ differs from *c*/*λ*_max_.

Thermal emission spectra are unlike spectra of discrete transitions, which have *ν_peak_* = *c*/*λ_peak_* with an intensity that is symmetric, or nearly so, about the characteristic frequency. Energy with a certain narrow frequency range is used to stimulate specific processes, e.g., laser light causes electronic transitions whereas sound waves cause low-frequency motions. However, heating a material requires redistributing the energy that is applied in some specified frequency range, which may be quite narrow, to the wide range of frequencies that comprise the thermal emissions of the material (Figure 2).

#### 2.1.2. Repercussions of Temperature Depending on Emitted Flux and Spectral Properties

Three facts derived from experiment and theory point to classical thermostatics incompletely describing solids:The hallmark of a hot dense body is that it emits heat over a wide spectral range (Figure 2). This unavoidable loss signifies that its state is dynamic, not static.Temperature governs the total flux emitted, with the following caveat:Because thermal emissions depend on the spectral properties of the material, *Q* may also depend on characteristics beyond the static physical properties considered in the historical model.

### 2.2. Connection of Steady-State Behavior with Coincident Adiabatic and Isothermal Conditions

Spherical geometries are conducive to examining total heat flux (Section 2.2.1). In contrast, Cartesian geometries are amenable for monitoring heat transfer across a solid (Section 2.2.2).

#### 2.2.1. Spherical Coordinates

Stefan–Boltzmann’s law, Equation (2), specifies a unique temperature for an object. Constant flux is implied: if the heat lost from a spherical object exceeds the energy input, the body cools, and conversely, if losses are retarded (e.g., via an insulating wrap), the body warms. In Stefan’s experiments, and in lightbulbs, electrical energy supplied at the center (Figure 3a) maintains surface output. For stars, interior nuclear fusion maintains a nearly constant outward flux. In these examples, flux from the much colder surroundings to the object can be neglected.

When rates of heat input at the center and output at the surface are the same, over any given time interval the amount of heat delivered and released is also identical: thus, conditions are adiabatic.

Isothermal conditions are commonly depicted as constant *T* over some significant expanse of space. However, because heat flow is ever-present per Equation (2) and different materials conduct heat at different rates, thermal gradients are unavoidable in a medium with finite size, per Equation (1). Boundary layers exist below spherical surfaces, since the object has both *T* and thermal conductivity that differ from those of the surroundings. For example, light from the sun originates in the photosphere (~600 km thick), which constitutes a boundary layer, being miniscule compared to the solar radius. Nearly grey emissions in the cavity experiments of Wein and Coblentz arise from the graphite coating, because their glass substrates have peaks in the infrared region, but are transparent at higher frequencies; see Figure 2.

In the laboratory, an apparatus (hot surroundings) provides finite flux into the material (ℑ_surroundings_ = ℑ_in_: Figure 3a). Steady state requires:ℑ_in_ = ℑ_out_ = constant(10)

At any moment, heat in = heat out, and so conditions are adiabatic. However, conditions are also isothermal because the temperature profile remains static in time and space. Specifically, at any given point (center, surface, or in between), some constant *T* is measured. Hence, thin spherical shells inside the body are isothermal. Likewise, the average *T* of the body is constant under steady state. Furthermore, its thermal gradient can be very small if ℑ_in_ and *κ* are low, thus approaching large regions of constant *T*.

Radial heat flow in a cylinder behaves like the sphere. The key difference is that the source would be a line, not a point.

#### 2.2.2. Longitudinal Flow in Cylindrical Geometry and in Cartesian Systems

To investigate behavior inside a solid, heat transfer experiments use geometries where both input and output are measured or controlled. Longitudinal flow (Figure 3b) is commonly used as this is one-dimensional and is described by Cartesian coordinates, even if the object is cylindrical. Boundary conditions exist: this paper follows Fourier, who treated these as distinct from conditions inside the material.

During steady-state conditions, the source and sinks of heat at the ends balance, so Equation (10) applies, and conditions are adiabatic. Furthermore, the heat flux is constant through any slice perpendicular to the thermal gradient, and the latter does not change with time, so the temperature in each perpendicular slice is likewise constant. However, because the source is at one end, and the loss is at the other, a thermal gradient exists from *T*_source_ at *x* = 0 to *T*_sink_ at *x* = *L*. It is immaterial whether the flux is radiatively applied (as in laser-flash analysis, LFA) used to measure *D*) or is supplied by electrical heating, or by contact with a hot plate. This equivalence has been amply demonstrated by benchmarking LFA against conventional heat transport measurements of metals, e.g., [15].

High *κ* and small ℑ produce shallow gradients, and so the limiting case of the whole body being a single temperature is approachable. However, because heat is emitted at any finite temperature, Equation (2), the gradient is never identically zero everywhere.

### 2.3. Equations of State, Elastic Behavior, and Work

The EOS is encapsulated as f(*V*,*P*,*T*) = 0, where f is some function. Behavior of *V* along each of the *P* and *T* axes provide important constraints. For simplicity, equations for isotropic solids are presented here. Importantly, f maps out the equilibrium behavior of a material, but contains no information on the processes of expansion or contraction.

#### 2.3.1. Classical Definitions and Their Link to Mathematical Constraints

One key physical parameter in the EOS is thermal expansivity:(11)$$\alpha_{P}\equiv \frac{1}{V}{\frac{\partial V}{\partial T}|}_{P}\;.$$

The *T* dependence of α is specific to any given material. For an isotropic substance, linear expansivity is 1/3rd of the volumetric expansivity, defined in Equation (11). 

Another key parameter is compressibility:(12)$$\beta_{T}\equiv -\frac{1}{V}{\frac{\partial V}{\partial P}|}_{T}=\frac{1}{\rho }{\frac{\partial \rho }{\partial P}|}_{T}=\frac{1}{B_{T}},\;$$
where *B_T_* is the bulk modulus. Its *P* dependence is likewise specific to the material of interest. Their second-order cross-derivatives are interdependent:(13)$${\frac{\partial \alpha }{\partial P}|}_{T}={-\frac{\partial \beta }{\partial T}|}_{P}={\frac{1}{B_{T}^{2}}\frac{\partial B_{T}}{\partial T}|}_{P}.$$

A convenient dimensionless parameter, known as the 2nd Grüneisen parameter, stems from Equation (13):(14)$$\delta_{T}\equiv -\frac{B_{T}}{\alpha_{P}}{\frac{\partial \alpha }{\partial P}|}_{T}=-{\frac{1}{\alpha_{P}B_{T}^{}}\frac{\partial B_{T}}{\partial T}|}_{P}.$$

The final important EOS relationship is obtained by setting *dV* = 0 in the mathematical identity:(15)$$dV={\frac{\partial V}{\partial P}|}_{T}dP+{\frac{\partial V}{\partial T}|}_{P}dT,$$
which gives the so-called thermal pressure:(16)$${\frac{\partial P}{\partial T}|}_{V}=-{\frac{\partial V}{\partial T}|}_{P}/{\frac{\partial V}{\partial P}|}_{T}=\frac{\alpha_{P}}{\beta_{T}}=\alpha_{P}B_{T}.$$

Actually, Equation (16) describes an isochore. Similarly, setting *d**P* = 0 in Equation (15) makes *α_P_* the relevant parameter, whereas setting *dT* = 0 in Equation (15) makes *B_T_* the defining property. Thus, Equations (11) and (12) describe behavior along an isobar and isotherm, respectively. The above equations constitute the EOS of a material.

Importantly, Equation (16) is identical to:(17)$$\left({\frac{\partial P}{\partial V}|}_{T}\right)\left({\frac{\partial V}{\partial T}|}_{P}\right)\left({\frac{\partial T}{\partial P}|}_{V}\right)=-1.$$

Any set of three variables can be manipulated in this manner, which stems from formulae analogous to Equation (15). Sets of four variables cannot be constrained solely through this approach: additional considerations are required. Those relevant to solids are covered next and in Section 2.4 on heat.

#### 2.3.2. Rigidity and Its Relationship to EOS Formulations for Solids

The special energy reservoir of solids, rigidity, provides their shape and strength (Figure 1). Rigidity permits a solid to remain motionless, except for the small, cyclical excursions of its vibrating atoms, while sustaining temperatures up to melting. In contrast, fluids flow under any stress, whereas some minimum stress (the elastic limit) must be exceeded for a solid to permanently deform below its melting temperature, e.g., [9].

Equations (11) to (17), currently considered to fully constitute the EOS, are valid for not only solids, but also liquids and gases. Completely describing a solid further requires establishing the dependence of *G* on *P* and *T*. 

Shear and bulk moduli determined from elasticity studies, which commonly use acoustic (subscript aco) methods and ultrasonic pulses [16], are defined as:(18)$$G=\frac{shear\;stress}{stear\;strain};B_{aco}=\frac{volumetric\;stress}{volumetric\;strain}$$

Rigidity and shear waves are only present in solids whereby shear deformation does not change volume. Hence, *G*, unlike *B_T_* or *B_aco_*, is not tied to heat. For this reason, the shear velocities are unrelated to the thermal Grüneisen parameter [17] which connects *B_T_* with *B_aco_* in the historical model (Section 2.3.6).

Elasticity is also represented by Poisson’s ratio (*μ*) and Young’s modulus (Ξ), where we do not use the conventional symbol *E* because it represents internal energy in classical thermodynamics. This pair is defined as:(19)$$\Xi =\frac{longitudinal\;stress}{longitudinal\;strain};\mu =\frac{lateral(transverse)\;strain}{longitudinal\;strain}.$$

The directional dependence of Equation (19) is obvious, and underlies our focus on isotropic solids. Note that *B*, *G*, and Ξ all have units of pressure, whereas *μ* is dimensionless.

The elasticity matrix, a 2nd order tensor [18] (p. 96), simplifies to three elements for isotropic solids: *c*_11_, *c*_44_, and the off-diagonal element *c*_12_. Because only three parameters are needed for isotropic solids, the elastic moduli are related:(20)$$\Xi =\frac{9BG}{3B+G}=2G\left(1+\mu \right)=3B\left(1-2\mu \right)\;\mathrm{where}\;\mu =\frac{3B-2G}{6B+2G}.$$

Although bulk properties can be represented by Equation (20), microscopic behavior being directional in anisotropic solids requires some approximations to provide *B* and *G* from measurements of such grainy material.

#### 2.3.3. Irrelevance of Friction to a Static Model and Implications for Work-Heat Relations

A plastically deforming solid evolves non-negligible frictional heat at some rate which then leaves the material at another rate. Inelastic processes depend on time: during such dissipative behavior, the material changes irreversibly, and restoration is impossible without additional energy. Detailed time-dependent models specific to the given situation are needed. Elastic materials evolve small amounts of heat [19], which constitutes a perturbation. It is not possible for such materials to indefinitely propagate compression waves as these will slowly be turned to heat. Similarly, compression and expansion are not truly reversible. As such, elastic materials, as defined by the material science and engineering communities, actually experience small amounts of inelasticity, and will require additional energy to offset losses to heat. The proportion requires assumptions beyond our static model, so it is not discussed further. Here, our use of “inelastic” and “elastic” differs subtly from materials science; in materials science, elastic materials are defined as ones which return to their original shape after deformation; instead, we use the original definition from physics whereby “elastic” indicates that all energy is recovered.

Two hundred years ago, Count Rumford’s cannon-boring experiments showed that work produces heat. His dissipative experiments involved time and friction. Mass was lost as well. As time is involved in Rumford’s experiment, changes in the cannon and the bore cannot be directly evaluated without rate laws.

Thus, the equivalence of work and heat explored historically is not assumed in our steady-state model of solids. Rather, elastic energy pertains to work (Section 2.3.7).

#### 2.3.4. Connection of the EOS with Perfectly Frictionless Elastic Behavior

An EOS describes the relationship between *P*, *V*, and *T* of a specified mass of a substance. A unique amount of heat energy and internal elastic energy is associated with any particular set of *P*,*V*,*T* coordinates, i.e., with any particular state. Nevertheless, the EOS does not by itself define what the latter quantities are: to determine those, knowledge of material properties is required. For a solid, a key component of the necessary information is the rigidity, yet rigidity is immaterial for gas.

Containment of the mass in some *V* for a given phase at any *P* or *T* is completely described by a reference state (*V*_0_ or *ρ*_0_), plus knowledge of *α*(*T*), *B*(*P*), and either cross-derivative (Section 2.3.1). Features of perfectly frictionless elastic solids (PFES) are summarized as follows:The perfectly frictionless elastic approximation is static: time is not involved and systems are fully restorable. That is, the ideal system is reversible (Figure 4b), although in a real system changes are made via manipulating and changing the surroundings.Because reversibility of the system and an instantaneous response to changing conditions are central to the PFES approximation, adding heat to the system has no effect other than raising temperature, after which *P* and/or *V* respond, in accord with imposed experimental constraints and the EOS. The time-dependent nature of heat uptake (Section 2.3.5) explains why this is the driver of change.Independence of mass and heat (Figure 4) and conservative behavior require separate treatment of variables related to mass occupying space (i.e., the EOS and shear modulus, *G*, which governs shape) and to heat occupying space (i.e., the heat content *Q*, storativity *C*, or a specific heat). Yet, the latter three parameters may depend on the size of the box (*V*), and thus on *P* (or *T*) conditions, as well as on *B* (or *α*) which describe volumetric changes.

#### 2.3.5. Uptake of Heat during Frictionless Elastic Behavior

Matter can be energized in various ways. We discuss application of a light pulse because light is pure energy, plus many types of experiments use light pulses.

Solids contain vibrating atoms. Their collisions must be nearly elastic to avoid large losses of heat. Ideal, harmonic oscillations meet this criterion, and are consistent with the PFES idealization.

These motions store energy in the solid over temperatures commonly accessed in laboratories, as described by the famous models of Debye and Einstein. Irradiating the material stimulates transitions among optical modes, where the dipole moment of a vibrating pair of ions changes (Figure 4c), discussed further in Section 4. Moreover, the pulse must penetrate the material. This distance is known as the skin depth, and can be inferred from optical properties: Wooten [20] provides a general discussion; Criss and Hofmeister [21] cover femtosecond spectroscopyof metals.

Since light propagates at a certain speed, uptake takes some finite time. Reaching equilibrium after the perturbation takes additional time, as the energy needs to be distributed among various vibrational modes that are connected with a higher temperature (Section 2.1). Thus, elevating *T* precedes adjustment of *V* or *P* to the new state. 

From the above, addition of heat involves three processes:

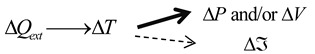
(21)

The present section concerns containment of mass (the heavy arrow), whereas Section 2.1 covers emissions (the dashed arrow). Section 2.4 focusses on heat (the LHS).

#### 2.3.6. Why Rigid Solids under Steady State Have One Bulk Modulus

Steady-state heat transport across a rigid solid is both adiabatic and isothermal (Section 2.2.2), a condition not addressed by classical theory. For an elastic solid in steady state, each incremental slice along the thermal gradient must have both constant *Q* and constant *T*. Therefore:(22)$${\frac{\partial V}{\partial P}|}_{Q}={\frac{\partial V}{\partial P}|}_{T}.$$

Hence, the isothermal bulk modulus (Equation (12)) of an elastic solid equals its adiabatic bulk modulus, denoted *B_S_* in the historic model, where *S* is entropy. This equality is not true for gases, due to their lack of rigidity combined with heat being carried by the molecules during their translational motions.

Elasticity experiments perturb a solid, which responds by propagating these perturbations internally as the form of waves. The response of the solid is then measured. Elastic waves have a well-defined frequency, whereas heat has a wide range of frequencies (Section 2.1). For a wave or pulse to heat a solid, the energy in the acoustic modes must be redistributed over a very wide frequency range, i.e., among the optic modes, overtones, combinations, and the continuum (Figure 4c). The process of redistribution and warming requires some finite time, and is not part of the measured, initial response of the solid, nor with the EOS. Moreover, not all exchanges are allowed. The special two shear (transverse) waves and one compression (longitudinal) wave are equivalent to the three acoustic modes of a crystalline solid. Acoustic modes are purely translational, where the atoms move in the same direction, whereas optical modes involve atoms in opposing directions [22]. For a vibrational mode of a crystal to directly absorb light, which includes heat applied to the solid, its dipole moment must change during the vibration [23]. This behavior is connected with symmetry and finite frequencies of optical modes at the Brillouin zone center, where acoustic modes have no energy: see [22] or [24] for examples and further discussion of the fundamental differences between acoustic and optical modes of crystal lattices. Regarding heat conduction inside the solid, data on the temperature and length-scale dependence of thermal diffusivity show that the process is largely diffusion of infrared light [3,7,25].

Because acoustic waves are not heat, elasticity experiments are nearly isothermal and also approximately adiabatic. Ultrasonic pulse methods [16] are popular, which supply less energy than a continuous wave. The key requirement is lack of frictional heating, which is reasonably accurate for stiff material. Classically, results of elasticity studies are denoted as *B_S_*. Since *S* is defined as *Q*/*T* in reversible experiments, *Bs* is referred to as the adiabatic bulk modulus. Because heat is irrelevant to elasticity experiments, we instead use the notation *B_aco_*, for acoustic bulk modulus, when referring to such data in Section 3.

Our model indicates *B_aco_* = *B_T_*, contrary to much literature, which posits that:*B_aco_* = *B_T_*(1 + *αγ_th_T*), historic.(23)
where the thermal Grüneisen parameter is historically defined as:(24)$$\gamma_{th}=\frac{\alpha_{P}B_{T}V}{c_{V}}=\frac{\alpha_{P}B_{aco}V}{c_{P}},\;\mathrm{historic}.$$

The historic difference in Equation (23) thus strongly depends on *α*. Like historic Equation (6), responses to *P* are cast in terms of responses to *T*, which is questionable.

#### 2.3.7. Young’s Modulus and Work in a PFES

How *V* responds to changes in *T* or *P* is described by thermal expansivity or the bulk modulus (Section 2.3.1). These physical properties are independent of path and of the process bringing about the change. Compressing a solid by external application of pressure, ideally hydrostatic, yields *V*(*P*) and the bulk modulus. In this case, work is performed by an apparatus, and heating is avoided to the fullest extent possible, so:(25)dQ=dT=0; to ascertain V as a function of P alone.

Conversely, determining *V*(*T*) and thermal expansivity requires changing *T*, while holding *P* constant. However, unlike *P*, which can be directly altered or controlled, changing *T* requires an intermediary step, i.e., applying heat and waiting for it to diffuse. Figure 4c illustrates the microscopic process of converting heat input to temperature. To expand the solid requires work. In the PFES idealization, an incremental addition of heat goes entirely into work:(26)dQ=dW=PdV; to ascertain V as a function of T alone.

If the addition is truly incremental, conditions remain approximately in steady state. The work performed expands the interatomic bonds. Resistance to this change is only partially governed by the bulk modulus, since solids also possess shear strength: see Meyers and Chawla ([18] Section 4.2) for discussion of Frenkel’s theory for shear strength. We use Young’s modulus (Section 2.3.2) to describe the resistance of the solid to incremental expansion as this is a measure of both *B* and *G*, and was used by Orowan to represent tensile strength ([18] Section 7.2). Section 2.4 explores the effect of the elastic energy reservoir of solids on their heat uptake.

### 2.4. Behavior of Heat in Perfectly Frictionless Elastic Solids during Steady-State Conduction

Density (*ρ* = *M*/*V*) describes how many atoms and molecules fill any given space. Analogously, heat density (*ε* = *Q*/*V*) describes how much heat occupies the same space (Figure 4a). Based on Stefan–Boltzmann’s law, which shows that the emissions (heat departing) from a volume *V* only depend on *T*, we deduce that for a PFES:(27)Q(T,P)=ε(T)V(T,P).

The function *ε* concerns only heat-energy since *T* is related to thermal emissions. Because light cannot compress, heat cannot compress.

#### 2.4.1. Specific Heat Definitions

Experiments do not measure *Q* directly, but rather record the response of matter to incremental energy augmentation. Measurements of heat capacity consist of perturbing steady state. Conserving mass makes specific heat germane, which is defined in terms of the heat externally supplied in order to raise a unit mass of some material by one degree:(28)$$c_{P}\equiv \frac{1}{M}{\frac{\Delta Q_{ext}}{\Delta T}|}_{P}={\frac{1}{\rho V}\frac{\partial Q_{ext}}{\partial T}|}_{P}.$$

Constant *P* is used in laboratory studies of solids. Heat capacity is similar to the above but is computed on a per mole basis. Multiplying Equation (28) by *ρ* gives storativity.

If volume is held constant:(29)$$c_{V}\equiv \frac{1}{M}{\frac{\Delta Q_{ext}}{\Delta T}|}_{V}={\frac{1}{\rho V}\frac{\partial Q_{ext}}{\partial T}|}_{V}.$$

Because *c_V_* data for solids are lacking, we focus on *c_P_*.

#### 2.4.2. Incremental Responses for a PFES

Equation (28) implicitly assumes that all applied heat goes into raising the temperature infinitesimally. Otherwise, the problem is insoluble. Moreover, this assumption is compatible with EOS formations and perfect elasticity (negligible dissipation). Hence:(30)$$\Delta Q_{ext}=\Delta Q_{int}=\Delta Q_{elastic}=\Delta Q\;\mathrm{or}\;\partial Q=\partial Q_{ext}.$$

The subscript ext on *Q* is hereafter discarded.

#### 2.4.3. Pressure Derivatives of Specific Heat during Steady State

For a reference point, the effect of compression on mass is null from mass conservation:


(31)
$${\frac{1}{M}\frac{\partial M}{\partial P}|}_{T}={\frac{\partial \left(\rho V\right)}{\partial P}|}_{T}={\frac{1}{\rho }\frac{\partial \rho }{\partial P}|}_{T}+{\frac{1}{V}\frac{\partial V}{\partial P}|}_{T}={\frac{1}{\rho }\frac{\partial \rho }{\partial P}|}_{T}-\frac{1}{B_{T}}=0$$


As discussed above, *ε* does not depend on *P*. Hence:(32)$${\frac{1}{Q}\frac{\partial Q}{\partial P}|}_{T}={\frac{\partial \left(\varepsilon V\right)}{\partial P}|}_{T}={\frac{1}{\varepsilon }\frac{\partial \varepsilon }{\partial P}|}_{T}+{\frac{1}{V}\frac{\partial V}{\partial P}|}_{T}={\frac{1}{\varepsilon }\frac{\partial \varepsilon }{\partial P}|}_{T}-\frac{1}{B_{T}}=-\frac{1}{B_{T}},\;\mathrm{for}\;\varepsilon \neq \varepsilon (P).$$

Taking the *P* derivative of Equation (28) gives:(33)$${\frac{1}{c_{P}}\frac{\partial c_{P}}{\partial P}|}_{T}=\frac{1}{Mc_{P}}{\frac{\partial }{\partial P}\left({\frac{\partial Q}{\partial T}|}_{P}\right)|}_{T}=\frac{1}{Mc_{P}}{\frac{\partial }{\partial T}\left({\frac{\partial Q}{\partial P}|}_{T}\right)|}_{P}=-\frac{1}{Mc_{P}}{\frac{\partial }{\partial T}\left(\frac{Q}{B_{T}}\right)|}_{P}.$$

Using Equation (32) leads to:(34)$${\frac{1}{c_{P}}\frac{\partial c_{P}}{\partial P}|}_{T}=\frac{-1}{B_{T}}\left\{1-\frac{Q}{Mc_{P}}\frac{1}{B_{T}^{}}\frac{\partial B_{T}}{\partial T}\right\}=-\frac{1}{B_{T}}+\frac{Q}{Mc_{P}}\frac{\partial \alpha }{\partial P}\approx -\frac{1}{B_{T}}.$$

The far RHS utilizes *d**B*/*d**T* ~0.001*B_T_* and the high *T* case where *c_P_* is nearly constant, which reduces Equation (28) to:(35)$$c_{P}MT\approx Q$$

The term with *Q* in Equation (34) is small from ~250 to ~1000 K, which covers experimental conditions commonly explored.

#### 2.4.4. Pressure Derivatives of Storativity during Steady State

Heat transfer experiments explore changes in storativity upon compression:(36)$${\frac{1}{C}\frac{\partial C}{\partial P}|}_{T,ℑ}={\frac{1}{\rho c_{P}}\frac{\partial \left(\rho c_{P}\right)}{\partial P}|}_{T,ℑ}=\frac{1}{B_{T}}-\frac{1}{B_{T}}+\frac{V}{Mc_{P}}\left\{\alpha \frac{\partial \varepsilon }{\partial P}+\frac{\partial^{2}\varepsilon }{\partial T\partial P}+\frac{\varepsilon }{B_{T}^{2}}\frac{\partial B_{T}}{\partial T}\right\}=\frac{1}{C}\left\{\alpha \frac{\partial \varepsilon }{\partial P}+\frac{\partial^{2}\varepsilon }{\partial T\partial P}+\varepsilon \frac{\partial \alpha }{\partial P}\right\}.$$

Because *C* already accounts for the box size, ∂(ln*C*)/∂*P* depends primarily on heat density. However, since heat density does not depend on *P*, then:(37)$${\frac{1}{C}\frac{\partial C}{\partial P}|}_{T,ℑ}={\frac{1}{\rho c_{P}}\frac{\partial \left(\rho c_{P}\right)}{\partial P}|}_{T,ℑ}=\frac{\varepsilon }{C}\frac{\partial \alpha }{\partial P}=\frac{Q}{CV}\frac{\partial \alpha }{\partial P}\leq 0,\;\mathrm{for}\;\varepsilon \neq \varepsilon (P).$$

The resulting negative sign for storativity requires that heat be shed during compression. From Equations (34) and (35), the magnitude is small.

The historic Equation (6) for *c_P_* leads to a strong dependence of *C* on *B_T_*:(38)$${\frac{1}{C}\frac{\partial C}{\partial P}|}_{T,ℑ}=\frac{1}{B_{T}}+\frac{1}{c_{P}}\frac{\partial c_{P}}{\partial P}=\frac{1}{B_{T}}-\frac{TV}{c_{P}}\left(\alpha^{2}+\frac{\partial \alpha }{\partial T}\right)=\frac{1}{B_{T}}\left[1-\frac{T\gamma_{th}}{\left(1-\alpha \gamma T\right)}\left(\alpha +\frac{1}{\alpha }\frac{\partial \alpha }{\partial T}\right)\right]≃\frac{1}{B_{T}},\;\mathrm{historic}\;\mathrm{model}.$$

#### 2.4.5. Temperature Derivative of Specific Heat during Steady State from Stefan’s Law

Taking the temperature derivative of Equation (28) and following steps similar to the above yields:(39)$$\frac{1}{c_{P}}{\frac{\partial c_{P}}{\partial T}|}_{P}=\alpha +\left[\varepsilon \frac{\partial \alpha }{\partial T}+\alpha \frac{\partial \varepsilon }{\partial T}+\frac{\partial^{2}\varepsilon }{\partial T^{2}}\right]/\left[\alpha \varepsilon +\frac{\partial \varepsilon }{\partial T}\right].$$

Importantly, greybodies are described by a unique temperature which is simply proportional to a characteristic frequency (Section 2.1). From Equation (9), the energy associated with the thermal emissions (light departing) from a solid is:(40)$$\;h\nu_{peak}=w_{3}k_{B}T\Rightarrow \;\mathrm{heat}\;\mathrm{energy}\propto k_{B}T.$$

Peak values, averages, and total energy involve different constants, but are all proportional to Boltzmann’s constant times *T* [26]. Because emission measurements providing Equation (40) were made at temperatures similar to the highest *T* reached in calorimetric and volumetric studies, neglecting the second *T* derivative of ε in Equation (39) is reasonable.

The denominator in Equation (39) can be recast as:(41)$$\varepsilon \left[\alpha +\frac{1}{\varepsilon }\frac{\partial \varepsilon }{\partial T}\right]=\eta \left[\frac{1}{V}\frac{\partial V}{\partial T}+\frac{1}{\varepsilon }\frac{\partial \varepsilon }{\partial T}\right].$$

As discussed earlier, adding heat makes the solid warmer and expands the solid. Expansion and increased temperature have opposite effects on ε. In lieu of complexities, such as bond bending in certain materials, *V* will not experience antagonistic effects. Thus, volumetric changes dominate the denominator, and the series expansion of Equation (39) becomes:(42)$$\frac{1}{c_{P}}{\frac{\partial c_{P}}{\partial T}|}_{P}≅\alpha +\frac{1}{\alpha }\frac{\partial \alpha }{\partial T}+\frac{1}{\varepsilon }\frac{\partial \varepsilon }{\partial T}=\frac{1}{\alpha }\frac{\partial \alpha }{\partial T}+\alpha +\frac{1}{\varepsilon }\frac{\partial \varepsilon }{\partial T},$$
where the far RHS lists the terms in order of size. Since α is about 0.01 times its logarithmic derivative at moderate to high *T*, whereas at low *T* the logarithmic derivative blows up, the α term in Equation (42) can be neglected. From the above, the heat density term is inconsequential at laboratory temperatures commonly used to measure *c_P_* and *V*. Hence, to a high degree of accuracy, the solution to Equation (42) and thus to Equation (39) provides a new equation:(43)$$\alpha (T)≅c_{1}(T)c_{p}\left(T\right).$$

Previous work compared averaged experimental values of α and *c_P_* and found equality at low *T* but a linear dependence at high *T* [27,28,29,30]. Bodryakov and colleagues [27,28,29,30] explained the discontinuous behavior on the basis of vibrations being the main energy reservoir in a solid, and did not consider elastic energy. Our derivation of Equation (43) suggests continuous behavior, but we have not yet incorporated the rigidity of solids.

#### 2.4.6. Heat Uptake Provides Non-Dissipative Work

Equation (43) is written to emphasize that the volume of a solid changes in response to heat uptake (Figure 4c). Thus, the parameter c_1_(*T*) describes the process of thermal expansion. When *T* is low, the solid is stiff because the bond lengths are small and bonding is strong. As *T* rises, the bonds lengthen and weaken. At high *T*, with weaker bonding, the same increment of *Q* added as at low *T* should cause greater expansion. Clearly, the structure of the solid should affect the function c_1_.

Basically, the applied heat does work. Using Equation (28) gives:(44)$$\;c_{P}M\Delta T=\Delta Q=work=PdV=F\Delta L,$$
where *F* is the force needed to expand the bond with length *L*. Young’s modulus (Ξ) represents the strength of the solid. The bulk modulus is not appropriate because it represents the change in *V* (or *L*) due to hydrostatic compression, thereby neglecting that solids may shear.

We begin with *F* = Ξ × area, and consider a spherical volume about an atom:(45)$$c_{P}M\approx \Xi 4\pi L^{2}\frac{\Delta L}{\Delta T}=\Xi 4\pi L^{3}\frac{\Delta L}{L\Delta T}=\Xi V\alpha ;\;\frac{\alpha }{c_{P}}\approx \frac{\rho }{\Xi }.$$

However, Equation (45) does not account for solids having a variety of structures with different bonding arrangements.

The properties α, *c_P_*, and *ρ* describe the bulk solid, so the structure is immaterial to these measurable quantities. The desired quantity, *F*, is related to Ξ, the number of atoms, and the number of bonds around each atom (i.e., atomic coordination of the structure). For example, diatomics have 2 atoms which share 1 bond, so *F* is proportional to Ξ/2. The same holds for the monatomic diamond structure, for which each atom is bonded to 4 others, mutually. Monatomics with the bcc structure have 2 atoms in the unit cell, which are bonded to 8 others, which double counts the bonds: thus *F* is proportional to 2Ξ/4. The 4 metal atoms in an fcc unit cell have 12 nearest neighbors, again double counting, so *F* is proportional to 4Ξ/6. Corundum has Al cations which are 6-coordinated, so Ξ/3 describes the force per cation. For the polyatomics with multiple sites, and given the above assumption of spherical atoms, *F* is estimated as being proportional to Ξ times the number of cations (*N*) divided by the number of atoms in the formula unit (*Z*):(46)αcP=ρΞNZ, for polyatomics.

From the examples listed above, Equation (46) also describes diatomic and monatomic solids. However, for monatomics, *N* is the number of cations in the unit cell, and *Z* is half the number of nearest neighbors in that unit cell.

#### 2.4.7. Ratio of Specific Heats

When volume is constant, heating the solid changes *P* in the interior:(47)$$\;\Delta Q=c_{V}M\Delta T=work=VdP.$$

Manipulating Equation (47) and using the definition of an isochore gives:(48)$$\;c_{V}M=V\alpha B.$$

The ratio is thus:(49)$$\frac{c_{P}}{c_{V}}=\frac{\Xi }{B}\frac{N}{Z}.$$

Structure pertains to the ratio because the interior forces composing Ξ differ from exterior application of pressure. This result cannot be tested as *c_V_* is not measured for solids.

## 3. Evaluation of New and Old Formulations via Comparison with Experimental Data

We evaluate whether available data support our model (Table 1) or the historical equations. We utilize compilations of data to decipher random errors. Studies of many materials by a single research group are another focus to reduce the effects of systematic uncertainties in comparisons.

### 3.1. Comparison of Bulk Moduli from Acoustic and Volumetric Studies

#### 3.1.1. Techniques

We use “volumetric” to include several experimental approaches that are conventionally considered to provide isothermal bulk moduli. X-ray diffractometry (XRD) and related techniques measure spacing of atomic planes, yielding unit cell volumes, whereas length-change measurements (e.g., [31]) measure macroscopic sample dimensions. Experiments are conducted at set points, presuming attainment of quasi-equilibrium at each step. The apparatus must supply a constant heat input to maintain constant *T*, while avoiding generation of extra heat from friction between moving parts.

Compression data are mostly collected at ambient temperature (NTP) rather than at 0 °C (STP). Ascertaining the effect of *P* on hard solids such as oxides is challenging because very high pressure is needed to induce substantial changes in *V*. Use of simple fits to describe *V*(*P*) data has become uncommon, perhaps because of erroneous statements that polynomial fits set ∂*B_T_*/∂*P* to 0 at *P* = 0 [32]. Rather, values for instantaneous derivatives depend on the accuracy with which *V* and *P* are measured, the spacing in *P* between data acquisition points, and the absence of deformation.

Commonly, volumetric data are fit to an assumed EOS. Popular forms assume that two constant values, namely the initial (*B_T_*_,0_) and 1st order derivative (*B*′ = ∂*B_T_*/∂*P*), suffice to delineate *V*(*P*). Large ranges in pressure are needed to establish the latter parameter, because it is the 2nd order pressure derivative of *V*. Additionally, uncertainties increase with *P*. Hence, *B*′ = 4 is commonly assumed. Although applying a certain form for the EOS is useful for comparisons, this approach introduces uncertainties by restricting parameter space. Convolution of *B_T_*_,0_ with *B*′ in EOS fits is a mathematical consequence of using only these two coefficients.

A different class of experiments determines elastic constants by recording the short-term response of materials to propagating waves or pulses [33]. The basis is equations relating stress to strain. Bulk and shear moduli are then calculated in accord with the symmetry of the structure and whether longitudinal or transverse waves are applied. Uncertainties stem from losses due to imperfect bonding of sample to transducer, imperfect orientation of single crystals, and use of approximate formulae for polycrystals. Spectroscopic methods e.g., Brillouin scattering are also well-established [34], but have similar limitations. However, since *B* is determined directly at ambient conditions, there is no need to assume an EOS. The term “acoustic” is used below to cover elasticity studies.

#### 3.1.2. Bulk Moduli for Solids at NTP

Compiled data on metals (Figure 5a) should be accurate because metals are fairly compressible and duplicate measurements exist. For example, Ledbetter [35] summarized measurements of zinc elastic constants presented in 11 studies, demonstrated consistency, and provided a tightly constrained average for zinc’s bulk modulus. Individual studies were sought when a metal was only present in either the elasticity database of Guinan and Steinberg [36] or in the XRD database [32], but not in both. We omitted any shockwave and XRD results that were included in the elasticity compilation.

Figure 5a shows that historical Equation (23) predicts that bulk moduli obtained from volumetric studies should be 1.6% lower, on average, than *B_aco_*. The calculated difference depends strongly on α-values near NTP, which are well-constrained for metals [37] and fairly large. Although the historical correction term of 1.6% is close to the experimental uncertainty in bulk moduli for individual metals, it is larger than the uncertainty of 0.5% of the fit for these 36 metals (see insets in Figure 5a). On average, the historical correction is unnecessary.

Bulk moduli values for electrical insulators and semiconducting Si scatter about the fit (Figure 5b). Within experimental uncertainty, *B_aco_* = *B_T_*. Applying historic Equation (23) to *B_aco_* predicts that bulk moduli should be only 0.6% lower than the trend in the data: this correction term is small because silicates and oxides have low α. Incompressible diamond (elemental C) and stishovite (SiO_2_ with the rutile structure) greatly influence the fit. Because α is low for insulators, little difference exists between data and the historic prediction, Equation (23).
Figure 5Comparison of data on bulk modulus from compilations of data from different experimental techniques. The x-axes depict XRD results from [32]: (**a**) metallic elements. Elasticity data (color points and line) mostly from [36]; supplemented by data on Pb and In [38] and Zn [35]. Calculations use *γ_th_* from [36]; recommended values of α from [37]; and *c_P_* from [39]; (**b**) electrical insulators and the non-metallic elements Si and C. Elasticity data on Si from [40]: otherwise from [41]. Additional XRD data, e.g., on BaF_2_, from [42,43,44]. Calculations use *γ_th_* and α from [45].
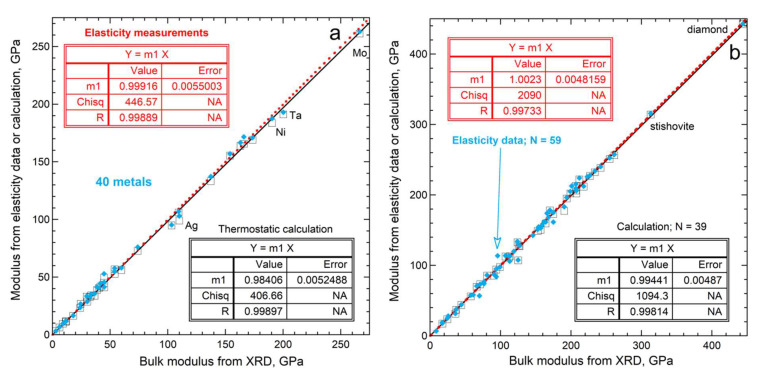


Statistical analysis provides further insights. Figure 6 shows histograms of the data in Figure 5. Many insulators have *B_aco_* < *B_T_* from volumetric studies (Figure 6a), which is the opposite of the historic predictions, Equation (23). For metals, *B_aco_
* tends to be slightly larger, whereas combining all data from Figure 5 provides an average difference very close to zero. Symmetry of the profile about a negligible difference (Figure 6a) points to a statistical origin for differences in bulk moduli measured on the same material with different techniques.

The historic predicted difference in bulk moduli, Equation (23), for metals is ~2× larger than that of insulators (Figure 6b), suggesting that acoustic and volumetric determinations should differ more for metals than for insulators. In contrast, Figure 5 and Figure 6a show that the differences between measured values of *B_T_* and *B_aco_* at NTP are smaller for metals than for insulators. These findings underscore that differences in bulk moduli at NTP for the ~100 samples in the compilations, many of which were measured multiple times, are caused by experimental uncertainties. Figure 5 and Figure 6 support our model.

#### 3.1.3. Uncertainty in Bulk Moduli Arising from Fitting Volume vs. Pressure

Bulk moduli are extracted by fitting *V*(*P*) to various polynomials or EOS formulae. To investigate the effects of fitting choices and measurement intervals (spacing of data points with *P*) we explore: (1) results for the metal Pb, which has pure samples due to its low melting point, and has been studied multiple times by many researchers; and (2) length-change measurements on many elements made using the same apparatus with similar procedures.

Figure 7 shows all metals and semi-metals for which both length-change and acoustic data exist. Vaidya et al. [46] made multiple runs of many samples. Their tabulated volumes, which may have been smoothed, were fit by us to 2nd order polynomials. Our results are similar to the polynomial fits of [31]. EOS parameters were averaged if multiple values were reported.

Results from both polynomial and EOS fits linearly correlate with *B_aco_* with a slope of unity (Figure 7a). Substantial differences exist in ∂*B*/∂*P* for the two types of fits [31] (their Table 5). As shown below for lead, a 3rd order polynomial is needed, but *P* = 4.5 GPa is insufficient to constrain curvature for most metals. This is underscored by measurements of tungsten [47] for which *V* depends linearly on *P*. Thus, using an EOS for W is an inaccurate representation. Discrepancies in Figure 7a for *B* > 130 GPa are attributed to both curvature in *V*(*P*) being too small for accurate fitting at high *B*, and also the trend of being highly influenced by uncertain *B* of incompressible W.

Figure 7b compares compressibilities, where the fitting is influenced most by the softest samples, rather than by the hardest. A 1:1 correlation exists, if the four softest samples are omitted. Fitting *V*(*P*) for Rb and K (not shown) required 5th order polynomials to account for inflection points, a behavior that is inconsistent with available EOS formulae. Apparently, Rb and K deformed in the tests. Accurate fits to Na and Se volumes required 3rd order polynomials. However, volumes for hard metals measured up to 4.5 GPa lack sufficient curvature to constrain a 3rd order polynomial fit. Thus, the four softest metals cannot be compared to the others in a consistent manner.

Thus, bulk moduli obtained from volumetric measurements equal the acoustic determinations, if *V* is measured and analyzed consistently. Notably, acoustic measurements also have experimental uncertainties and most metals studied are polycrystalline, for which elasticity formula (i.e., the Voigt–Ruess–Hill formulation) is approximate (Section 3.1). Such effects cause the scatter in Figure 7.

Volumetric data on Pb from four studies are fit with a 3rd order polynomial (Figure 8a), providing *B_T_*_,0_ = 45.5 ± 0.5 GPa. Results from Schulte and Holzapfel [48] are not included because a table of volumes was not presented and resolution of the points on their figures was insufficient for accurate digitization. They applied a two-parameter EOS to their own and previous data, yielding *B* = 42 ± 5 GPa with individual studies ranging from 39 to 51 GPa. All fits cluster about 40 to 42 GPa. Figure 8b omits this average because shockwave data were included by [48]. We excluded fits to both fcc and bcc phases.

Various approaches to fitting volumes obtained at 298 K give a wide range of values for *B_T_*_,0_. A key factor is the maximum pressure obtained. When the full stability field for lead is used, EOS fits with two parameters, give lower values for *B*_0_ than fits to a 3rd order polynomial, which uses three parameters. The constraint of *V*/*V*_0_ = 1 is not included in the free-parameter count, as this is fixed in all approaches.

Compressing Pb to 8.6 GPa is not sufficient to accurately establish curvature (Figure 8a). The very high *P* studies have widely spaced points, which limit the accuracy of fitting. Regarding two-parameter polynomial fits, these can give higher or lower B than either the 3rd order polynomial or the EOS, depending on several factors. From the fitting in Figure 8a, the comparison in Figure 8b, and considering variations among the data sets, we infer that accurately determining *B_T_*_,0_ requires meeting several conditions: dense spacing of points, volumetric data over a wide range of pressures, accurate (or at least consistent) determination of pressures, and using a fit with three parameters or more (in addition to *V*_0_).

Notably, use of a 3rd order polynomial is consistent with anharmonic oscillations. Further exploration of polynomial fitting to extensive data sets is needed, but is beyond the scope of the present report.

Bulk moduli at NTP obtained from volumetric data even for Pb, which is a fairly soft metal, include substantial uncertainties. For hard substances, uncertainties are larger, which explains differences in scatter in Figure 5a,b. Figure 5, Figure 6, Figure 7 and Figure 8 indicate that elasticity measurements record isothermal bulk moduli.

#### 3.1.4. Comparison of Acoustic to XRD Determinations of ∂B/∂T for Solids

Comparison of elasticity and XRD data on bulk modulus at high temperature is limited because few substances have been measured at high *T* with both approaches. Challenges arise from large thermal gradients in the material and/or apparatus. We focus on accurate measurements of soft solids, as these have large α which permits definitive evaluation. Alkali halides, alkali metals, and lead data meet these criteria. Due to experimental uncertainties, *B_aco_* does not always exactly equal *B_T_* at NTP (Section 3.1.1, Section 3.1.2 and Section 3.1.3). Therefore, we compare values of ∂*B*/∂*T*, which has a negative sign.

Our model (Section 2.3) requires that values of ∂*B*/∂*T* are the same for acoustic and volumetric determinations. In contrast, the historic Equation (23) leads to:(50)$$\;{\frac{\partial B_{T}}{\partial T}|}_{P}=\frac{1}{\left(1+\alpha \gamma T\right)}\frac{\partial B_{aco}}{\partial T}+\frac{-B_{aco}}{{\left(1+\alpha \gamma T\right)}^{2}}\left[\alpha \gamma +\alpha \frac{\partial \gamma }{\partial T}T+\frac{\partial \alpha }{\partial T}T\right],\;\mathrm{historical}.$$

Below ~2000 K, the derivatives on the RHS are smaller than the product *αγ*, as shown in the tables in Anderson and Isaak [54], which include hard oxides and soft alkali halides. The two derivative terms furthest to the right are similar in magnitude but opposite in sign. For *T* accessed in experiments, Equation (50) is reasonably represented by:(51)$$\;{\frac{\partial B_{T}}{\partial T}|}_{P}\sim \frac{\partial B_{aco}}{\partial T}-\alpha \gamma B_{aco},\;\mathrm{historical}.$$

The terms on the RHS are similar in magnitude [54]. Since ∂*B_T_*/∂*T* is negative, volumetric measurements should give a stronger response to *T* than elasticity measurements.

Yagi [55] determined volumes of four alkali halides to 9 GPa and 1073 K in a piston-cylinder apparatus using XRD. NaCl was included with each sample to provide an internal pressure scale, where Decker’s [56] calibration was used. Bulk moduli (Figure 9) were extracted using the Murnaghan two-parameter EOS, and were found to agree with those from length-change measurements [57]. Mismatch occurs with acoustic determinations at any given *T*, but *B* vs. *T* curves from volumetric and acoustic studies are parallel. The only exceptions (Figure 9) are from studies that disagree with subsequent measurements. In addition, acoustic determinations by various authors on each sample differ by varying amounts at 298 K. Within experimental uncertainty, equivalence of the derivatives from acoustic and volumetric techniques is confirmed.

Historic Equation (51) gives 8.3%K^−1^ for CsCl which is larger than, but similar to, ∂*B_aco_*/∂*T* = 5%K^−1^ (Figure 9). For LiF and NaF, Equation (51) gives 4.8 and 4.2%K^−1^, respectively, which are smaller than ∂*B_aco_*/∂*T* = −10.6 and −6.9%K^−1^, respectively (Figure 9). Yagi’s [55] measurements of volumes provided similar ∂*B_T_*/∂*T*, rather than values about half the size of ∂*B_aco_*/∂*T*. The historic model is not supported.

Regarding lead (Figure 8b), volumetric data of Strässle et al. [51], analyzed using Skelton et al.’s [60] adaptation of Decker’s [56] scale, gave *B*(*T*) parallel to the trend of the cryogenic acoustic data. This finding is irrespective of using an EOS or a polynomial fit to *V*(*P*). Strässle et al. [51] were puzzled by their EOS determination for *B_T_* at 298 K, with the EOS being as predicted by historic Equation (23), but not their 80 K value, and so reevaluated their data with an untested cryogenic calibration, attributed to in a personal communication, which yielded the desired historic result. As shown in Figure 8a, the EOS analysis of lead volumes at low *P* underestimates the bulk modulus, so their fitting approach only appears to agree with this historic adjustment. Rather, fitting lead volumes over the stability range of its bcc phase to a high-order polynomial agreement with *B_aco_*, and do not require amending via Equation (23). As demonstrated for the alkali halides, bulk moduli trends with *T* for lead from volumetric and acoustic techniques are parallel, and so the historic correction is refuted.

Soft alkali metals have also been studied by both XRD and acoustic techniques (Figure 10). The trends are nearly parallel. At 298 K, length-change measurements better agree with *B_aco_* than with the cryogenic volumetric studies, except for Na. The historic correction at 298 K exceeds or matches the difference between the various measurements, and thus agreement of absolute values involves random experimental uncertainties as is evident from compiled data (Figure 5, Figure 6 and Figure 7).

### 3.2. Response of Heat Capacity at NTP to Compression

Compressing a solid affects specific heat and storativity in different ways, permitting two independent evaluations using Equations (34) and (37). The historic Equations (6) and (38) differ considerably from our model, providing two additional tests.

Two types of measurements exist for specific heat of solids as a function of pressure near ambient temperature. Calorimetric measurements have been performed on 3 metals (Section 3.2.1), whereas transport measurements involve 20 insulators, plus 3 metals by difference (Section 3.2.2). Only for Cu and MgO do multiple *c_P_*(*P*) measurements exist.

#### 3.2.1. Static Compression Techniques

Metal wires were studied at pressure using electrical heating, where a correction term was applied to account for thermal losses. This term involves resistivity of the wire and is larger (for Cu) or similar (Ni, Al) in magnitude to uncorrected ∂ln(*c_P_*)/∂*P* [65] (their Figure 7) and [66] (their Figure 3). Uncertainty for the reported value is substantial and cannot be less than ~10% uncertainty for the change in resistivity with *P*, e.g., [67].

#### 3.2.2. Dynamic Compression Techniques

Measurements of transport properties as a function of pressure provide ∂ln(*c_P_*)/∂*P* in two different ways. First, from Equation (5):(52)$$\frac{\partial \ln\left(\kappa \right)}{\partial P}=\frac{\partial \ln\left(\rho \right)}{\partial P}+\frac{\partial \ln\left(c_{P}\right)}{\partial P}+\frac{\partial \ln\left(D\right)}{\partial P}=\frac{1}{B_{T}}+\frac{\partial \ln\left(c_{P}\right)}{\partial P}+\frac{\partial \ln\left(D\right)}{\partial P},$$

Different methods yield *κ* or *D*, and occasionally both properties. Combining results yields ∂ln(*c_P_*)/∂*P* by difference whereby uncertainties of the terms sum.

Only experiments on large (~ mm thickness) samples are considered, to permit comparison of the results, since transport properties linearly depend on length-scale at small *L* [7]. Dynamic measurements provide *D* and *κ* vs. *P* for three metals, MgO, and olivine (Figure 11). Uncertainties are roughly ±10% for each transport measurement, which makes ∂ln(*c_P_*)/∂*P* obtained by difference uncertain by ±20%. Figure 11 omits measurements of three samples: Gd melts very close to NTP; Zn has a hexagonal structure and the orientations differed in the *D* and κ experiments; whereas results on garnet gave positive ∂ln(*c_P_*)/∂*P*, which is unexpected, and is probably due to large uncertainties in small derivatives for this hard insulator.

Second, certain dynamic experiments on insulators simultaneously provide *κ* and *C* as a function of *P* (e.g., [68,69]). Alkali halides, Si, and MgO were explored (as detailed in Figure 11). If a graph for *C* was presented, we used the slope and *B_T_* to calculate ∂ln(*c_P_*)/∂*P* from the LHS of Equation (36) instead of the EOS approach as used by authors.

Most studies note high uncertainties. Nominal uncertainties at NTP of 5% for transport measurements are gauged by metal standards. Insulators have larger, systematic errors from contact loss and radiative transfer. However, their effect is reduced by comparing logarithmic derivatives.

**Figure 11 materials-15-02638-f011:**
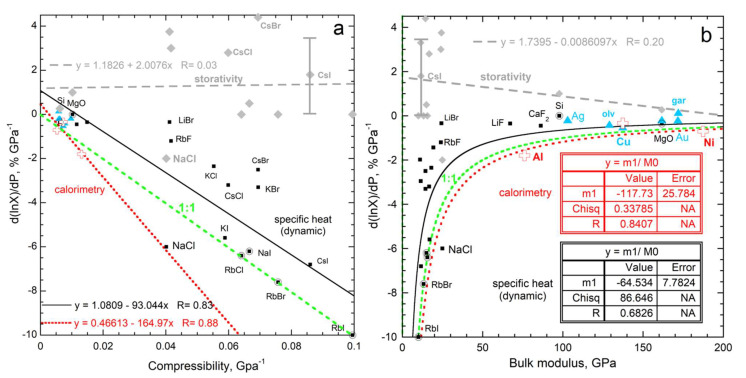
Graphs showing the response of storativity and *c_P_* to pressure: (**a**) dependence on the inverse of *B*; (**b**) Direct dependence on *B*. Grey diamonds and grey dashed line = directly determined storativity: sources = [70,71,72,73] where the error bar is from Gerlich and Andersson [70]. Black squares and solid line = specific heat from *C*, where circles = data where *C* did not discernably depend on pressure. Open cross and red dotted line = metal *c_P_* directly measured by calorimetry [65,66]. Aqua triangles = heat capacity obtained by difference (sources: [69,74,75,76,77,78]). Green short dashed line = ideal correspondence.

#### 3.2.3. Relationship of the Pressure Response of Specific Heat and Storativity to Bulk Moduli

Available data show that measured values of ∂ln(*C*)/∂*P* for insulators weakly depend on compressibility or the bulk modulus, as predicted by Equations (36) and (37). The results are scattered (Figure 11), rather than inversely depending on *B*, which disagrees with the historic Equation (38). Considering the large experimental uncertainty, storativity is independent of pressure. This explanation is supported by soft solids, which are prone to deformation, having *C* dependent on *P*, whereas the hard solids show little change. It is also consistent with the seemingly random variation in the sign of ∂*C*/∂*P*.

In contrast, ∂ln(*c_P_*)/∂*P* decreases roughly linearly with *B*^−1^ (Figure 11). Values for the slope vary with the technique (calorimetric or dynamic). The slope is uncertain, due to 10 to 20% uncertainties for the various approaches and the fact that an EOS is used to process storativity, which adds uncertainty—basically, this is also a difference approach. All data from all approaches combined (not shown) give a slope of about −1 or −100%. This slope is consistent with Equation (34), which shows that compression of the lattice controls the response. Within experimental uncertainty, the energy density is independent of pressure.

#### 3.2.4. Evaluation of the Historic Relationship of the Pressure Response of Specific Heat to Thermal Expansivity

Figure 12 evaluates historic Equation (6) using sources listed in [79] and Figure 5 and Figure 11. The temperature derivative of α is uncertain, and contributes to scatter. Measured ∂ln(*c_P_*)/∂*P*, on average, responds strongly to compression whereas the correlation with historic Equation (6) is poor. The existence of a rough link is attributable to compressible solids also having large α; see, e.g., Anderson and Isaak [54].

### 3.3. Connection of Thermal Expansion to Heat Uptake and Internal Strength

Uncertainties in the properties considered here increase with *T*. Uncertainty in density is negligible, compared to that of the others, which generally increases in the order *c_P_* < Ξ < α. Because thermal expansion is small and measured as a response to *T*, values are impacted by the measurement range and fitting procedures, parallel to the limitations in determining *B_T_* (Section 3.1.1).

#### 3.3.1. Ambient Temperature

Figure 13 compares the ratio α/*c_P_* to the ratio *ρ*/Ξ without considering effects of structure after Equation (45). Agreement is reasonable for the monatomic elements, but with considerable scatter. This could be due to ~25 elements having *N*/*Z* = 1, but being anisotropic, as discussed below. The correlation for insulators is linear, with a slope differing from unity predicted by Equation (45). Its value of nearly ½ is as expected from our structural analysis of the interatomic forces (Section 2.4.6).

Figure 14 evaluates the effect of structure on interatomic forces (Equation (46)). Semiconducting Si and Ge are omitted because these have negative thermal expansivity at low temperature (see Appendix A). Diamond is included with the insulators because its structure differs from the remaining solid elements, which are metals plus the semimetal Te. Figure 14 analyses the three different structures that describe most metallic elements. Data on the insulators and face-centered cubic (fcc) metals confirm Equation (46), whereas data on the body-centered cubic (bcc) and hexagonal close packed (hcp) metals require an additional factor. Discrepancies for the non-cubic solids, i.e., olivines, among the insulators and hcp metals point to anisotropy, which affects measurements of both α and Ξ, but was not accounted for in our analysis (Section 2.4.6). Corundum is hexagonal, but its physical properties such as thermal conductivity are nearly isotropic and so this behaves like the cubic insulators. For the remaining non-cubic structures, additional information is needed to describe their forces, so we do not pursue details of their behavior below.

Divergence of anisotropic samples from Equation (46) in Figure 14 suggests that shear (deformation) underlies mismatch since both affect the amount of longitudinal vs. lateral strain. DeJong et al. [83] modelled failure modes of four bcc metals. Their categorizations of ductile (shear) vs. brittle (tension) failure agree with available experimental data. Equation (46) overpredicts α/*c_P_* for ductile Nb and Ta but agrees with α/*c_P_* for brittle Mo and W. Shear being important means that some of the heat energy goes into deforming rather than solely expanding the lattice: consequently, α/*c_P_* is overestimated.

To quantify the effect of ductile behavior, the data in Figure 14 are recast as a difference and a ratio in Figure 15a,b, where each is compared to Poisson’s ratio (Section 2.3.2). The rigid insulators agree well with Equation (46), excluding the orthorhombic olivines. The scatter is otherwise attributed to experimental uncertainty, mostly in α, due to its small size (discussed further below).

As deformation becomes an increasingly important component of elasticity in each of the fcc and bcc metals, α/*c_P_* is increasingly overestimated by Equation (46). Positive discrepancies (overestimation of the energy supplied towards expansion) are associated with transverse strain being large compared to longitudinal strain. Thus, deformation accounts for departures of individual metals from the trends established for each of the fcc and bcc structures, but it does not account for their different trends.

One explanation of the different trends is that our use of structure to link interatomic forces to Young’s modulus is an oversimplification. Ionic-covalent bonds for the insulators are strong and electrons are localized, so assuming that forces are controlled by nearest-neighbor couplings is strongly supported. Bonding in metals involves delocalized electrons, so 2nd nearest neighbors participate somewhat in the force field around a cation. The fcc cations have 12 nearest neighbors at 0.707 *L* and six 2nd nearest neighbors at *L*. Because 2nd nearest neighbors are few and are at 1.4× longer distances, using 12 bonds is reasonable but low. Increasing bond number to 13.8 would provide a slope of unity in Figure 14a. If a bond count of 13.8 has been used in Figure 15, this would place most metals within uncertainty of exact agreement with Equation (46). These samples have typical μ = 0.2 to 0.33, which overlaps with the range of the insulators. For another estimate, an extended unit cell with 5 atoms would have 18 bonds (double counted), giving *Z*/*N* = 1.8 instead of 1.5. Agreement with Equation (46) for all fcc metals occurs midway between these estimates.

The bcc structure has eight nearest neighbors at 0.866 *L* and six 2nd nearest neighbors at *L*. Secondary bonding is more substantial than a perturbation. Considering an extended unit cell suggests *Z*/*N* = 7/3 = 2.33 instead of 2 for the primary bonds. This modified value does not explain the overall underestimation of expansion at ambient *T* caused by heat uptake by bcc metals. Further evaluation would require a close look at the original sources of data, particularly α. Experimental uncertainties may be a problem for the highly reactive alkali metals. This potential limitation is supported by the well-studied, non-reactive bcc metals (Fe, Mo, W) lying on the 1:1 line of Figure 14b, whereas Ta is slightly off, due to its high ductility, discussed above.

#### 3.3.2. Temperature from a Few Kelvins to Nearly Melting

Previous comparisons of α(*T*) to *c_P_*(*T*) averaged many data sets [27,28,29,30], which removes random errors. Because systematic errors also exist, we compare individual data sets in Figure 16a which should accurately represent each of α(*T*) and *c_P_*(*T*). Evaluating the temperature dependence of Equations (43), (45), or (46) further requires accurate data on Ξ(*T*). Fortunately, comparing rather few samples suffices because specific heat depends similarly on *T* for diverse materials, both simple (e.g., [24]) and complex [84]. Likewise, solids expand similarly as temperature climbs: for details, see Appendix A. Similar behavior of Ξ with *T* for different substances has also been observed (Figure 16b), leading to common use of the formula:(53)Ξ(T)=ΞT=0−aTexp(−bT),
where Ξ at the limit of 0 K as well as constants *a* and *b* are fitting parameters [85,86,87,88,89,90,91].

We focus on diverse cubic substances with multiple and accurate measurements over wide *T*-ranges. Pure substances, where disordering of cations among sites is negligible, are considered. Appendix A provides graphs comparing α to *c_P_* as a function of *T* for Al, Fe, Mo, Ta, Au, diamond, Si, MgO, Al_2_O_3_, Y_2_Al_3_O_12_, NaCl, and KCl.

The five metals examined in detail have a sufficient range of densities, Young’s modulus, and structures to permit the evaluation of our new equations. Rows 3 to 6 and columns VB, VIB, VIII, IB, and IIIA of the periodic table are represented. Figure 16 shows the ratio α/*c_P_* above 200 K, where data on Ξ exist. As *T* further increases, α increases more strongly with *T* than does *c_P_*, such that the proportionality factor *c*_1_ in Equation (43) grows non-linearly with *T* at very high *T*.

Semiconducting Si has negative α at low *T*, but behaves similarly to isostructural diamond at high *T* (Appendix A). Because α(*T*) being disconnected from *c_P_*(*T*) was also observed over the Curie point of Fe (Appendix A), we propose that heat energy goes into expanding the lattice when no other process exists that can uptake the increment applied. In Fe, the additional process is electromagnetic. Section 3.3.1 argued that deformation likewise diverted heat-energy from thermal expansion. From both observations, we suggest that the process in Si involves electronic state changes. This hypothesis could be tested against impurity content for Si and Ge.

Figure 16a shows that the ratio α/*c_P_* depends on *T*. Its derivative with *T* (the slope) depends on Ξ near 298 K, in accord with Equation (45). Trends are flat and similar for materials with very high Ξ. The slope steepens as Ξ decreases. Density and Young’s modulus together affect the low *T* intercept of α/*c_P_*. The behavior exhibited in Figure 16a supports the findings of Section 3.3.1.

The slopes of α/*c_P_* correlate reasonable well with ∂Ξ/∂*T* for diverse materials (cf. Figure 16a,b). Insulators include extremely tough diamond, three incompressible oxides with varying structural complexities, and two soft alkali halides. Bonding ranges from ionic to covalent. Bass’s [41] summary table shows that the *T* derivatives of elastic properties vary considerably among studies of the same material. Non-linearity of the response contributes. Hence, uncertainties in ∂Ξ/∂*T* are substantial. On this basis of large experimental uncertainties, and because density changes with *T* are even smaller, ambient *ρ* was considered in Figure 16b and Figure 17.

Figure 17 shows that thermal expansion of solids is more easily accomplished at high *T* because the solid gradually weakens with *T*. Shear is a substantial competing mechanism for Au and Ta, causing our model to overestimate expansion at room temperature (Figure 15) and above (Figure 17). Within experimental uncertainties, data at elevated temperature support our model for the response of strong solids to heat.

## 4. Discussion and Implications

We present a new model for static physical properties of solids for the case of steady-state heat flow. The classical “thermodynamic” model does not account for ubiquitous heat flow or for the dissimilar physical behaviors of gas and solids, encapsulated in Figure 1. In particular, the fact that heat-energy and mass move independently in a solid, unlike gas, and the quantitative description of heat flow by Fourier are neglected in classical theory. An equally significant omission was neglecting the constant emission of heat from a solid, as experimentally established by Stefan, and theoretically supported by Boltzmann’s derivation of the *T*^4^ dependence of flux.

Independent behavior of heat and mass in solids stems from their rigidity and strength: hence, elasticity is the dominant energy reservoir of solids (Table 2). Moreover, coherent transverse motions that embody two of the three acoustic modes in solids have no counterpart in gas. As elasticity is connected with interatomic forces within a solid, this reservoir involves potential energy (P.E.) and is distinct from heat storage, which is known to be kinetic energy (K.E.) from study of gases. The nature of heat storage is covered in Section 4.1.

Addressing these omissions led to relationships among the physical properties of solids that differ from the historical formulae. Table 1 lists key new formulae which we have evaluated with available data. Some additional results cannot be verified because no data exist for solids, for example, on *c_V_*. Testing many different solids required use of compilations, which introduced uncertainties. Nonetheless, available data show that for solids: Only one bulk modulus exists, so the historically alleged difference between acoustic and volumetric moduli is unsupported. Likewise, the isothermal and adiabatic values for the 2nd Grüneisen parameter (Equation (14)) must be identical.Changes in heat content with pressure are controlled by the compressibility, which dominates changes in specific heat at moderate laboratory temperatures.Changes in heat content with temperature are described by specific heat by definition. Specific heat and thermal expansivity are linked, as the process of increasing *V* involves overcoming the elastic, tensile forces within the solid. Deformation solely occurs as shape changes arising from shear stresses uptake energy without expansion, confirmed by comparison of results from Equation (46) to Poisson’s ratio for cubic solids. If heat stimulates other processes, expansion is reduced as in Fe, or even reversed, as in Si.

### 4.1. Heat Storage Reservoirs and Permissible Exchanges of Energy

All solids store heat. Those with multiple types of atoms have short-range vibrational motions that interact directly with light, as occurs in polyatomic gases. Applied light-energy is absorbed by these cyclical, small-scale motions, then communicated during equilibration (Figure 4c), and stored as heat.

For gases, the molecular vibration reservoir is in addition to that of the longer scale, translational motions, as is well-known. At equilibrium, these different energy reservoirs must have the same temperature. Partial temperatures do not exist. Yet, the heat-energy content associated with each reservoir need not be the same, and in fact is not. One example is diatomic gas, for which the translational K.E. reservoir is larger than the vibrational reservoir. Equal temperatures of reservoirs are in accord with the zeroth law and with Stefan’s observations: at equilibrium their heat losses (fluxes) must match.

Solids must behave similarly. Thus, monatomic solids which lack optical modes (e.g., bcc and fcc structures) must have some heat storage reservoir. These metals emit approximately as blackbodies and consequently absorb light at all frequencies. This optical continuum is thus the manifestation of the main heat storage reservoir in metals (Table 2). Continuous absorption is consistent with the wide range of distances, and thus dipole moments, between the moving, loosely bound electrons and the approximately stationary cations.

Energy cannot be freely exchanged among all reservoirs. Rules exist for energy exchange and in many cases prohibit it. Rules are evident from experiments. Acoustic modes in solids are not stimulated by light, even when its frequency matches that at the zone edge, because the dipole moment does not change during these coordinated motions of the cations [23]. Thus, neither the sole longitudinal acoustic mode nor the two transverse acoustic modes participate in emitting light. Without a flux, the acoustic modes have no temperature and so the elastic reservoir not being in equilibrium with the heat reservoir does not violate the zeroth law. Our discussion is in accord with the nearly free electron model [24]. Heat transfer is a disequilibrium phenomenon that is not relevant to the equilibrium state: for measurements and theoretical assessment of electronic and vibrational transport in metals, see Criss and Hofmeister [21].

From another perspective, the elastic reservoir maintains and even increases its energy as the 0 K limit is approached (Figure 8, Figure 9 and Figure 10). The acoustic modes have more energy as *T* decreases because the bonds become shorter. Solids become more rigid with decreasing *T*. Even polyatomic alkali halides behave in this way (Figure 9). If the elastic reservoir exchanged energy with the heat reservoir achieving low temperatures might be impossible. Moreover, acoustic waves propagate extremely long distances, for example, 1000s of km inside the Earth. Weak attenuation, unlike that during heat transfer which attenuates over ~ mm lengths, is only possible with negligible energy exchange. Exchange of energy between reservoirs is observed to occur only when the length scales associated with different energy inventories are similar. This restriction is a consequence of the Virial theorem of Clausius [92]. The entire solid sample responds elastically to stress, whereas the interactions of solids with heat and light are microscopic.

### 4.2. Key Variables

The essential thermodynamic variables that govern solids under steady state are mass, volume, temperature, and stress. Although mass is held constant in our model, *M* remains important because atomic constituents dictate structure and bonding, and therefore affect the interactions of the particular solid with stress and with applied heat. Because heat is never stationary, the supply of flux is crucial, but is assumed to equal the flow out, so the total energy content is independent of time in our model. That is, the constraints of steady-state dictate the relevant variables and how measurements are made. In more detail:

Heat storage and the solid’s response to applied heat are probed by perturbing the system, i.e., by monitoring the response of the solid to incremental heat additions (pulses) and recording this as a heat capacity. Temperature is actually a consequence of an influx of heat energy to the solid, which is maintained externally.

Stress has direction and can be separated into an isotropic component (hydrostatic pressure, *P*) which alters volume but not shape, and a deviatoric component, which alters shape but not volume. Elastic properties describe the changes (strain) in response to stress. For solids, bulk modulus (inverse of compressibility) has been the focus as this is the response to hydrostatic compression, and also occurs in gas. For a solid, its response to shear stress is equally important, but gases offer no resistance to shear. Moreover, steady state involves a direction of heat flow, and thus Young’s modulus and Poisson’s ratio better represent mechanistic responses during steady state. Because the elasticity reservoir is independent of the heat reservoir, thermal expansion is related to heat uptake through the rigidity of the solid and its directionality, including in anisotropic solids.

### 4.3. Reservoirs vs. Historic State Functions

The neglect of the huge reservoir of elastic energy in solids in the historical model requires revision of essential variables (Section 4.2) as well as of the associated energies, historically referred to as state functions. For solids, elastic energy replaces the state function denoted internal energy. Unlike internal energy, elastic energy is independent of temperature, flux, and heat.

From the definition of specific heat (Equation (28)), integration provides *Q*, the heat content. A constant of integration is unnecessary because at the limit of *T* = 0, flux also approaches the null limit. Otherwise, a substance could cool below absolute zero. The absence of flux at 0 K means that no heat is stored at this limit. Otherwise, a small amount would be emitted. Heat content replaces the historic enthalpy function for solids.

Entropy for a solid is related to its configurational disorder. Defining entropy in terms of *Q* and *T* is problematic because heat flow is ubiquitous. In the historic approach, *S* is a variable, yet enthalpy, i.e., *Q*, is a state function. Our model lacks this inconsistency.

The classically defined free energies of Gibbs and Helmholtz are not needed to describe solids. Rather, our analysis shows that only two very different types of energy exist in our ideal, time-independent solid. One reservoir consists of storage of elastic energy of the solid, which is potential energy since motions do not exist until the system is perturbed, i.e., activated by adding heat. The second type is heat content, which is kinetic energy, since atomic motions are always present at finite *T*, while taking on different forms (Table 2).

## 5. Conclusions

We constructed a new thermodynamic theory for the perfectly elastic frictionless solid that accounts for the vastly different physical character of solids and gas, while addressing the ubiquitous flow of heat. Our model shares two inherent limitations with the historic model, as it is also macroscopic and independent of time. Our model differs by: (1) considering steady-state conditions for heat flow, which are common and achievable; and (2) accounting for the rigidity of solids. The latter shows that the energy associated with their elasticity, which was ignored in classical models, is large and independent of their heat reservoir. Our focus on perfectly frictionless elastic solid is analogous to the classical model of the ideal gas: in both theories, exploring the limiting case of elastic, conservative behavior sets the stage for more complex, realistic behavior.

Our new equations, which differ substantially from historic ones, were confirmed using available data on isotropic solids. Although validation is limited to simple structures, all bonding types (metallic, covalent, and ionic) are represented and agree with our model, supporting its generality. We also demonstrated that counterpart equations in the historic model, which are based on behavior of gases and neglect rigidity, are not supported by the same data.

Incorporating elasticity into a thermostatic model reveals the mechanism for thermal expansion: namely, the added heat performs incremental work, which is required to transition between equilibrium states, but is opposed by the interatomic bonds that define the structure and rigidity of the solid. This link explains why the temperature dependence of α is complex. Other key equations (Table 1) provide simple relationships for the pressure responses of specific heat and heat content. The relationship between the two specific heats is simple: when cast as *Bc_P_* = Ξ*c_V_*, it is apparent that their difference lies in whether pressure is externally controlled, or whether the resistance to heating is internal to the solid. Furthermore, we show that isothermal and isentropic (adiabatic) compressibilities are identical, which is consistent with thermal expansivity taking on one value (isobaric) and isentropic conditions not being germane.

Many different disciplines apply various historic thermostatic relations to solids. Materials science and engineering fields should find our interrelationships among thermal expansivity, specific heat, and Young’s modulus useful in designing materials, because both strength and thermal response are germane to many applications. Geophysical research would greatly benefit from our new theory because the slowly varying, high-pressure and high-temperature conditions in Earth’s deep interior cannot be reached in the laboratory, and so the historical equations have been relied on. Substantial revisions are expected for the thermal structure of planetary interiors, since these bodies are very compressed and thus very strong solids.

## Figures and Tables

**Figure 1 materials-15-02638-f001:**
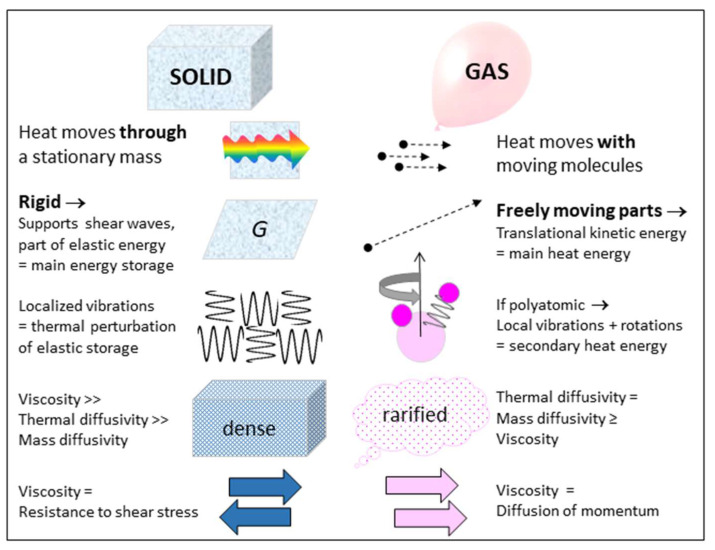
Summary and comparison of the characteristics of solids and gases most relevant to heat and its flow. The shear modulus, *G*, describes a special type of stored energy in solids, which is part of the elastic energy, the main reservoir. Atoms are shown as balls, with dotted arrows indicating direction of long-distance motions. Sine waves without arrowheads indicate local, back-and-forth, microscopic motions.

**Figure 3 materials-15-02638-f003:**
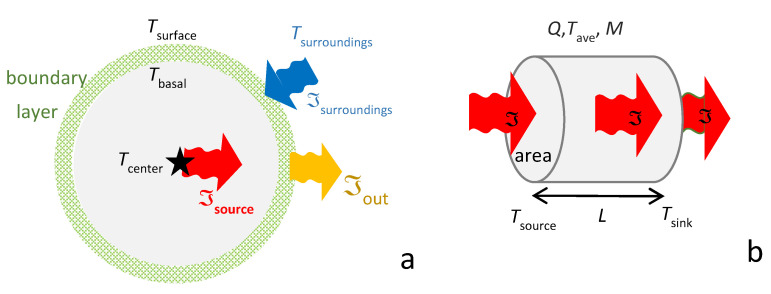
Schematics of conditions: (**a**) Spherical symmetry, which also applies to radial flow in a very long cylinder. Matter (grey circle) emits heat in accord with its temperature (orange squiggle arrow), but emissions are actually sampled from a surface boundary layer (stippled green shell). Constant flux is maintained either by a source (star) and/or externally (blue arrow); (**b**) Longitudinal flow in Cartesian (or cylindrical) symmetry. At steady state, flux along the special direction is a constant that is independent of position, so the axial thermal gradient is independent of time, and perpendicular slices are isothermal.

**Figure 4 materials-15-02638-f004:**
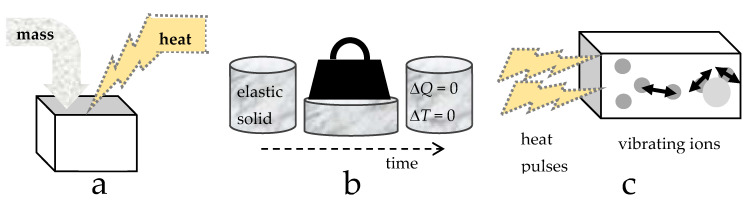
Schematics of an ideal, perfectly elastic solid: (**a**) any given volume can contain a quantity of mass, and can independently contain some quantity of heat-energy; this independence underlies our model; (**b**) Essence of elastic behavior. Squeezing (increasing pressure) changes *V*, and thus does *P*-*V* work, but does not generate heat so *T* is unchanged. Upon release of pressure, a perfectly elastic frictionless solid returns to its initial volume. See text for discussion of shear and shape changes; (**c**) Receipt of small amounts of heat by a PFES. Within a short, but finite, distance, the pulse encounters vibrating ions. When energy of the applied light matches some transition energy, the affected vibrations become excited, attaining a higher energy state (e.g., an overtone). Subsequent interchanges give an overall higher vibrational energy of the collection, which imparts a higher temperature. Both steps take time.

**Figure 6 materials-15-02638-f006:**
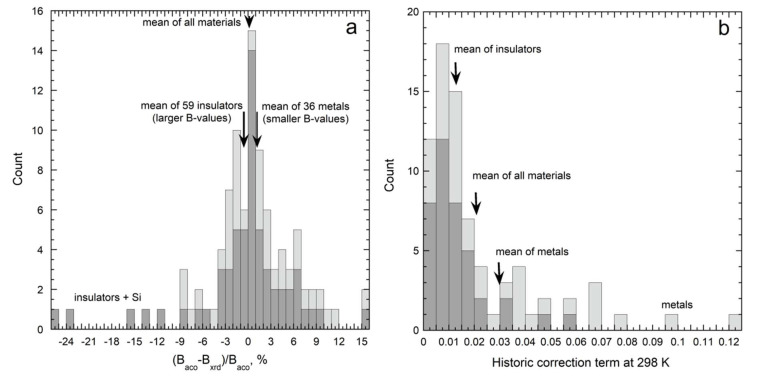
Statistical presentation of the data from compilations. See Figure 5 for literature sources. Light grey = metals; dark grey = insulators and Si. Arrows point to various mean values: (**a**) histogram of the difference between elasticity and volumetric measurements of bulk moduli, in percent; (**b**) histogram of the product *αγ**T* at 298 K. Expansivity data were found for 39 of the insulators that had both types of bulk moduli measurements.

**Figure 7 materials-15-02638-f007:**
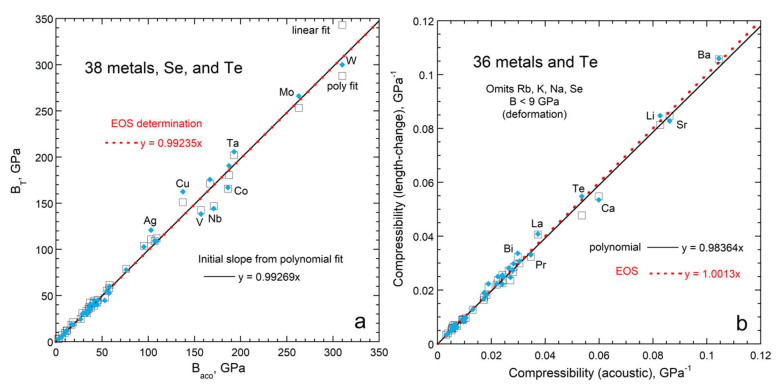
Comparison of different measures of metal bulk moduli. Length-change measurements from [31,46,47] were fit to EOS by the authors and to 2nd order polynomials here. Most acoustic determinations are from compilations listed in Figure 5. Vaidya and Kennedy [47] provide additional acoustic data: (**a**) direct comparison. Both linear and polynomial fits are fit for tungsten, because curvature in *V*(*P*) was not resolved. Gold was not measured, but the other noble metals have relatively large *B_T_*; (**b**) inverse comparison. The four softest metals were excluded because these not only needed 1–3 more terms for accurate fitting, but more importantly, the sigmoidal dependence of their *V* on *P* indicated deformation. We did not fit the initial slope because the lowest *P* data may be affected by slight deformation.

**Figure 8 materials-15-02638-f008:**
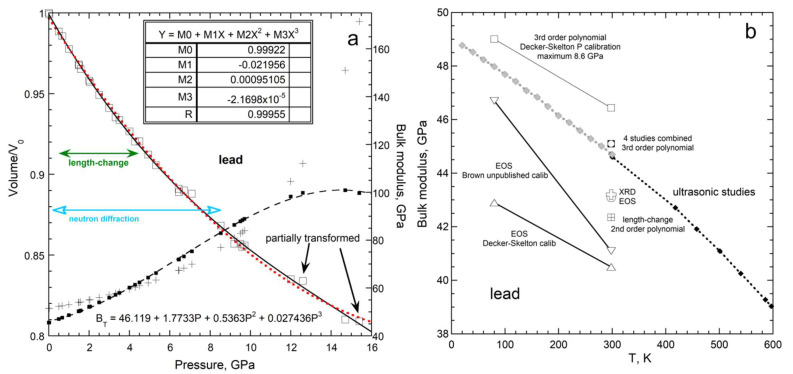
Lead volumes and bulk moduli, mostly from DAC studies: (**a**) polynomial fit combines results from [31,49,50,51]. Double arrows denote pressure ranges. XRD experiments probed the whole stability field (to 16 GPa) but with few data points. Dotted curve = the 2nd order fit to *V*, where + = the corresponding *B*(*P*). Inset lists the 3rd order fit to *V* vs. *P* (solid curve), with filled squares for the resulting *B*(*P*), which is fit to the listed 3rd order polynomial. This fit gives slightly higher initial B than calculation; (**b**) temperature dependence of bulk moduli. Diamonds = *B_aco_* (grey from [52]; black from [53]). Square in circle = result from panel a. Open squares and various triangles = several fits to neutron diffraction data [51], as labelled. Other open symbols = reported EOS values of [31,49,50,51].

**Figure 9 materials-15-02638-f009:**
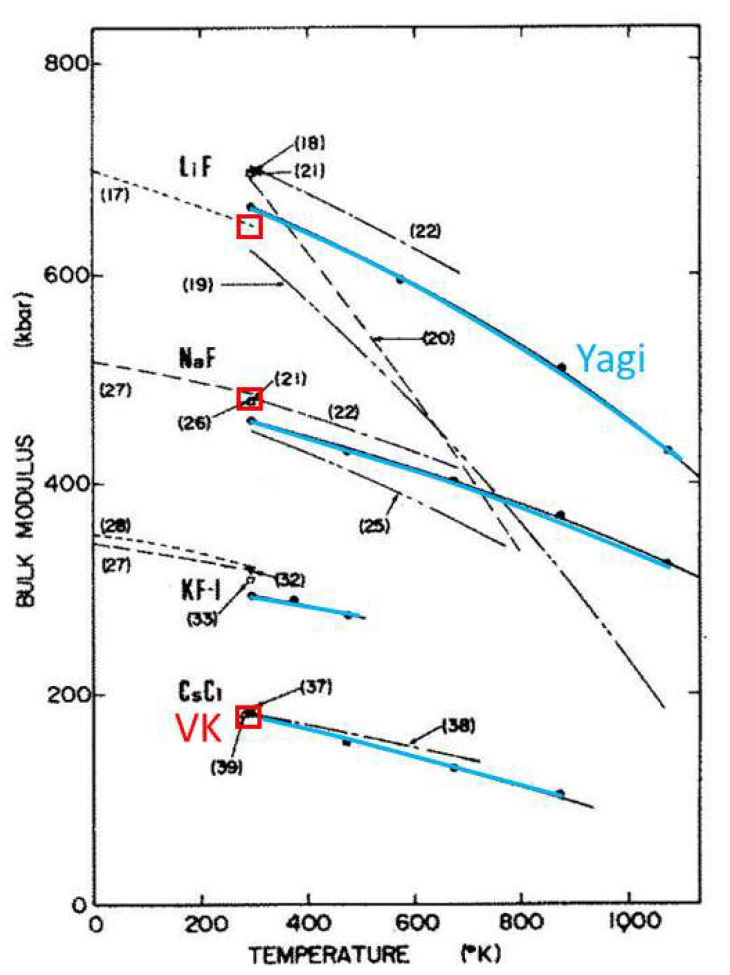
Bulk modulus of alkali halide as a function of temperature. Blue curves = EOS fits of Yagi [55] to his XRD data on LiF, NaF, the low-*P* B1 phase of KF, and CsCl with the B2 structure. Numbers in parentheses denote previous work cited by [55]. Red squares and “VK” = length change data [57], where too few data collections were made on KF to provide a reliable *B_T_*. Broken curves = acoustic data compiled by Yagi [55], where his references 19 and 20 are incompatible with other studies. For example, Hart [58] confirmed *B_aco_*(*T*) from curve 22 for NaCl, i.e., the work of Jones [59]. Modified after Yagi [55] (his Figure 8) with permission.

**Figure 10 materials-15-02638-f010:**
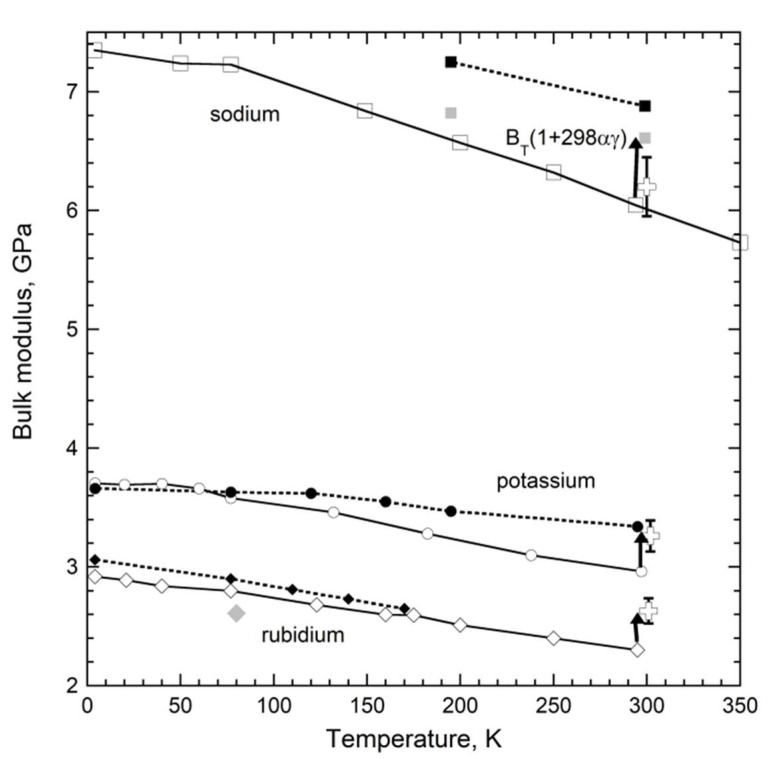
Temperature dependence of bulk moduli for alkali metals. Filled symbols = acoustic data of [61,62,63]; grey represents previous work cited therein. Open symbols = volumetric (XRD) studies analyzed using simple forms for the EOS [64]. Open cross = length-change data [46], which are closer to acoustic results than to *B* from XRD. Otherwise, squares show various data on Na; circles for K; and diamonds for Rb. Arrow at 298 K shows the historic Equation (23) applied to XRD data.

**Figure 12 materials-15-02638-f012:**
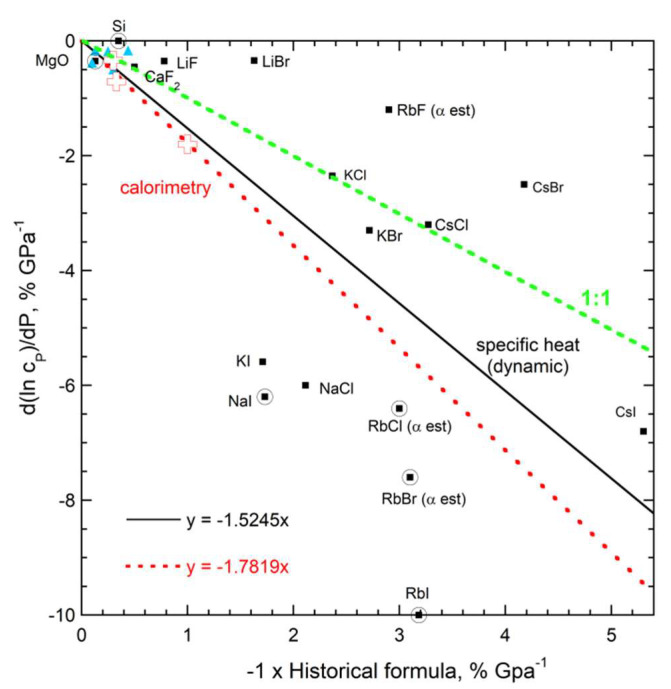
Comparison of the measured *P* response of specific heat to the thermostatic formula (6), which is peculiarly based on thermal expansivity describing compression. The difference method (blue triangles) provides a cluster of points, and so was not fit. Red = direct calorimetry measurements. Green dashed line = 1:1 correspondence, for reference. Circles = materials for which storativity was not discernably affected by compression. Open cross = metals, by calorimetry. Black line = fit to the scattered dynamic measurements. Data sources listed in Figure 11.

**Figure 13 materials-15-02638-f013:**
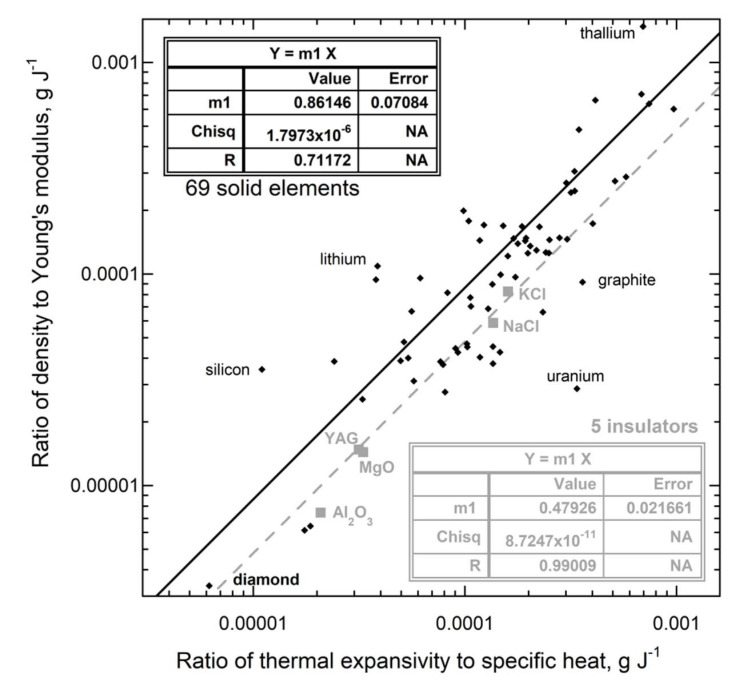
Evaluation of Equation (43). Data on α, *ρ*, and *c_P_* from [37,39,80,81]. Young’s modulus data from [41,82]. The five insulators are examined below: see Section 3.3.2 for details and sources.

**Figure 14 materials-15-02638-f014:**
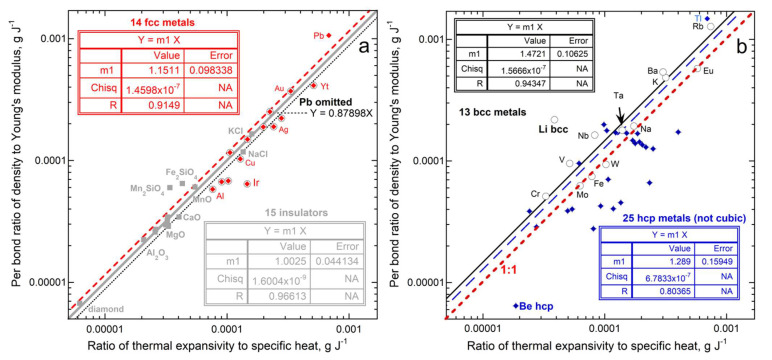
Dependence of α/*c_P_* on *ρ*/(Ξ*N*/*Z*). Literature sources of data on elements are in Figure 13. For the insulators, tables of [54] were used, where Co_2_SiO_4_ was omitted because α was estimated. Fits are least squares and are labeled with the number of solids in each category: (**a**) insulators and cubic fcc metals. Lead strongly influences the slope due to its softness, as shown by the two fits. Iridium has little influence as it is near a cluster of points. Orthorhombic Fe_2_SiO_4_ has a shearing transition whereas α for orthorhombic Mn_2_SiO_4_ is unconfirmed; (**b**) cubic bcc and hexagonal hcp metals. Outliers Li and Be have very small cations and few valance electrons.

**Figure 15 materials-15-02638-f015:**
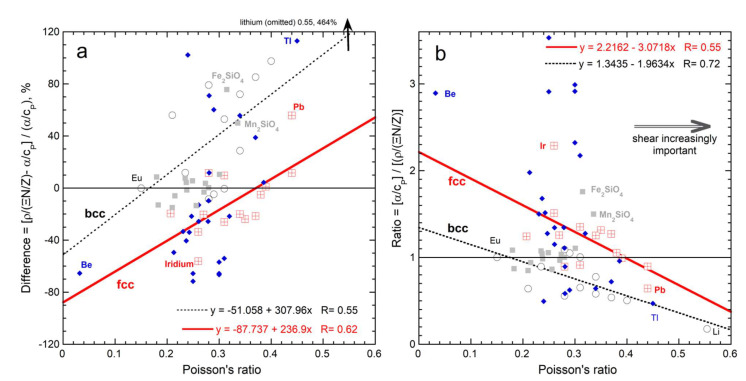
Measures of discrepancy of the data from (46) as a function of Poisson’s ratio. Data on μ from [41,82]; see Figure 14. Fine line = ideal match. Dotted line and circles = bcc. Thick line and squares = fcc. Diamonds = hcp: (**a**) difference = {*ρ*/(ΞN/Z) − α/*c_P_*}/(α/*c_P_*) in percent; (**b**) ratio of α/*c_P_* divided by *ρ*/(ΞN/Z).

**Figure 16 materials-15-02638-f016:**
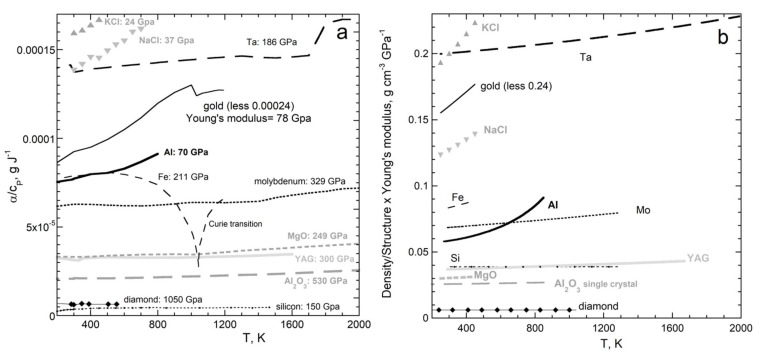
Evaluation of Equations (43) and (46) at high *T* for well-studied solids: (**a**) dependence of α/*c_P_* on temperature. See Appendix A and Figure 13 and Figure 14 for data sources. Jumps in Ta curve result from data-combining studies. The graph begins at 200 K as cryogenic data were previously shown to closely correspond [27,28,29,30]; (**b**) dependence of *ρ*/Ξ with the structural factor on *T*. Constant ambient *ρ* was used due to uncertainties in Young’s modulus. Measured data on Ξ from [85,86,87,88,89,90,91]. For Au, Fe, MgO, NaCl, and KCl, we used *T* derivatives near and above 298 K for *B* and *G* from [41] to compute dΞ/d*T*.

**Figure 17 materials-15-02638-f017:**
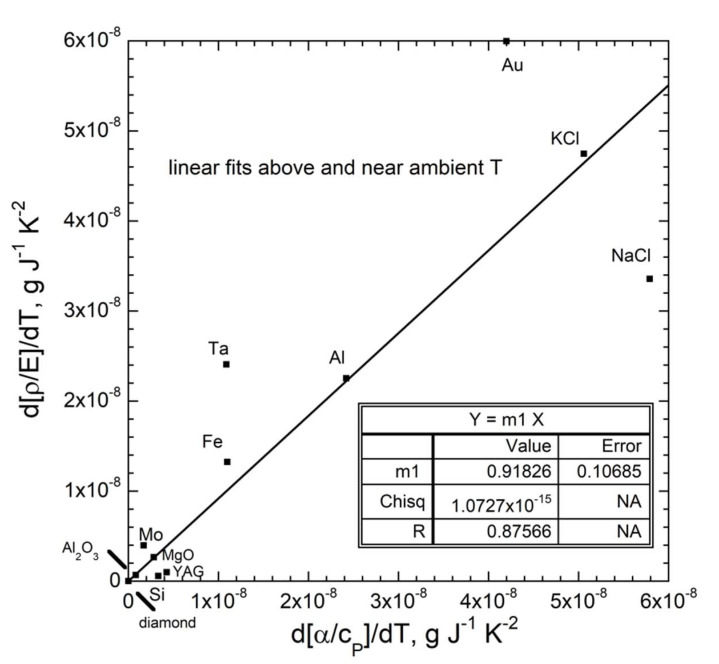
Comparison of the temperature dependence of the RHS and LHS of Equation (43). The effect of structure is not included. Data for these solids are described in Figure 16 and Appendix A.

**Table 1 materials-15-02638-t001:** Physical properties of perfectly frictionless elastic solids under steady-state heat flow.

New Formula	Theory	Experimental Confirmation
*B_T_* = *B* from elasticity measurements	Section 2.3.2 and Section 2.3.6	Section 3.1 (ambient and elevated *T*)
${\frac{1}{c_{P}}\frac{\partial c_{P}}{\partial P}|}_{T}≃-\frac{1}{B_{T}}\equiv \frac{1}{V}{\frac{\partial V}{\partial P}|}_{T}$	Section 2.4.2	Section 3.2 (ambient *T*)
$\frac{\alpha }{\rho c_{P}}\propto \frac{1}{\mathrm{Young}’s\;\mathrm{modulus}}$	Section 2.4.6	Section 3.3 (ambient and elevated *T*)

**Table 2 materials-15-02638-t002:** Dependence of energy reservoirs on the state of matter and the complexity of its atomic constituents.

Type	Motion	Solids		Gases	
		Manifestation	Energy Storage	Manifestation	Storage
monatomic	Displacements parallel to path	Longitudinal acoustic mode	Longitudinal stress/strain ^1^	Translational K.E.	Heat
	Displacements perpendicular to path	Transverse acoustic modes	Transverse stress/strain ^1^	n/a	n/a
	Electron-cation dipoles	Optical continuum	Heat	Collisions	n/a ^3^
polyatomic	Longitudinal	Longitudinal acoustic mode	Longitudinal stress/strain ^1^	Translational KE	Heat
	Transverse	Transverse acoustic modes	Transverse stress/strain ^1^	n/a	n/a
	Electron-cation dipoles	Optical continuum	Heat	Collisions	n/a ^3^
	Cyclical, tiny ^2^	Optical modes	Additional heat	Internal modes	Heat

^1^ For solids, these together compose elastic storage of energy in tension-compression and shear, respectively. ^2^ These internal motions and energies are in addition to those described for monatomics above, but are also found in certain monatomic structures such as Raman modes (diamond and hcp metals). Although Raman modes do not directly absorb light, their overtone/combinations do. ^3^ Presumed to be brief and conservative in the historical model.

## Data Availability

All data were previously published.

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
