# Peer review of "Thermodynamic Relationships for Perfectly Elastic Solids Undergoing Steady-State Heat Flow"

_materials, 2022, doi:10.3390/ma15072638_

Round 1
Reviewer 1 Report
In the paper entitled: " Thermodynamic Relationships for Perfectly Elastic Solids Undergoing Steady-State Heat Flow", the authors have proposed the new thermodynamic equations for the perfectly elastic frictionless solids.
After I read the paper and check some of the references, beside my research on thermo-elasticity I concluded that this paper has been good organized and I recommend to publish it.
The relation of the bulk module to elastic one and complexity of temperature dependency of the coefficient of thermal expansion are the main contribution which have not been considered in most of the previous thermo-elasticity theories.
In this paper they have constructed not only a new thermodynamic theory,
they have also compared the results of the other experimental papers (different metals)
to support their idea.
Their paper is also have a good literature review which can be considered
as a review paper in this field.
Their conclusion and data are reliable as you can see in other papers of these authors.
The level of this paper and its demonstrations are very high
and I recommend you to publish it without any hesitation.
During my research in the field of thermoelectricity for temperature dependent material properties,
I encounter their concern and the proposed equations will be helpful in material characteristics in micro to macro scales.
Author Response
Response: We thank the reviewer for his thoughtful and positive remarks. Like the reviewer, we also encountered inconsistencies in the literature, but in the different area of high pressure geophysics. We hope that our new equations will indeed be helpful to materials research.
The reviewer checked that presentation of results could be improved, but did not provide any specific suggestions. So we clarified some places and in particular made sure that the captions were clear. Several figures were improved
Reviewer 2 Report
Dear authors
Thank you for this very well organized and presented piece of work, I have few things that I suggest to be amended.
1) the purpose of the paper need to be summarized into few points rather than writing a whole paragraph.
2) the results are too long and contains too many equations and figures that I would rather prefer to be presented in less bulk.
3) limitation of the study need to be added to the paper.
Best Regards
Author Response
Response: We thank the reviewer for his thoughtful and positive remarks. Regarding specific comments:
1) We agree that the purpose was too long. So, we split this into two subsections. The discussion of gas vs solid behavior now has its own subheading “1.1 Different behaviors of solids and gases may affect thermostatic equations.” This includes figure 1, which is background material.
After figure 1, we inserted “Section 1.2 Purpose and limitations of the paper.” We reversed the order of the last two paragraphs, and added a sentence on limitations.
Although our revised section on purpose is not shorter, these changes better distinguish purpose from limitations, which are related, but distinct, aspects of a paper.
2) The results section is necessarily long, as it must be thorough. No other reviewer suggested that the number of figures or equations be reduced. We believe that all material we present is needed for a convincing validation of our model, so we did not reduce this section, but did mode several clarifications to the text. We improved figures 1,3,11,12, and 15 which should help with clarity.
3) We agree that we should specify the limitations of our new model. We tried to make these clear in our revised introduction (see point 1 above). We also added a discussion of model limitations to the first paragraph of the Conclusions, for completeness.
Reviewer 3 Report
The report by Hofmeister et al., entitled “Thermodynamic Relationships for Perfectly Elastic Solids 2 Undergoing Steady-State Heat Flow” describes about the steady state heat flow in various samples including the macroscopic structure. To apply their concepts, they assumed various parameters and hypothesis. I see that this work deals exclusively with the theoretical aspects and no such experimental points are covered. The Introduction section is covering the important aspects and is with proper thermodynamic models, theoretical description of solids conducting heat in a steady state with link of temperature to the heat reflux, adiabatic and isothermal conditions, spherical coordinates, cylindrical geometry, rigidity in solids, irrelevance of nature of friction in static models, and EOS connection to the frictionless elastic models are well covered. The study also discussing the points related to the heat uptake during the frictionless elastic behavior, importance of bulk modulus in steady state materials, heat in perfectly frictionless elastic solids during steady-state conduction, heat uptake in non-dissipative work. The authors have studied the new and old formulations by comparing with the experimental data. From the theoretical aspects and the assumptions that the authors taken into account to make their theory to be working, I recommend this work for the publication.
I have reviewed the manuscript and see no comments to raise.
I see that the manuscript is a theoretical work and they have taken some published content to prove that their theory is correct.
Therefore, the work can be acceptable in its present format without need of any additional formatting. The reason to give my review report as "accept after minor revision" is just to check whether there may be any plagiarism issues as the authors referred to the published work. Otherwise, everything is ok from my side.
Author Response
Response: We thank the reviewer for his thoughtful and positive remarks. Regarding his specific comments, we obtained permissions for figure 9. Otherwise the figures are new, and while most utilize data from multiple sources, those publications are listed in the captions.
Reviewer 4 Report
Anne M. Hofmeiste et.al manuscript titled "Thermodynamic Relationships for Perfectly Elastic Solids Undergoing Steady-State Heat Flow" derives thermodynamic relationships for perfectly frictionless elastic solid (PFES). In contrast to historical "thermostatics" the energetic reservoir "rigidity" in solid is accounted to derive thermodynamic relationships in the current manuscript. In addition mass-energy conservation, Fourier's and Stefan-Boltzmann' laws are applied to derive the relationship. The significance of new thermodynamic relationship derived would be highly promising for designing materials and also understand planetary interiors as pointed out by the author. I strongly recommend for publication considering significance of new thermodynamic relationship derived for PFES.
Author Response
Response: We thank the reviewer for his thoughtful and positive remarks.
He noted that the introduction could be improved, but did not make specific comments. We think the changes to address Rev. #2 provide the needed improvements. These changes are:
We split the purpose into two subsections. The comparison of gas vs solid behavior now has its own heading “1.1 Different behaviors of solids and gases may affect thermostatic equations.” This includes figure 1, which is background material.
After figure 1, we inserted “Section 1.2 Purpose and limitations of the paper.” We reversed the order of the last two paragraphs, and added a sentence on limitations.
Although the section on purpose is not shorter, these changes better distinguish purpose from limitations, which are related, but distinct, aspects of this paper.